# VisualPRM400K: An Effective Dataset for Training Multimodal Process Reward Models

**Weiyun Wang**[1,2], **Zhangwei Gao**[3,2], **Lianjie Chen**[4,2], **Zhe Chen**[5,2], **Jinguo Zhu**[2],
**Xiangyu Zhao**[3,2], **Yangzhou Liu**[5,2], **Yue Cao**[5,2], **Shenglong Ye**[2], **Xizhou Zhu**[4,2],
**Lewei Lu**[7], **Haodong Duan**[2], **Yu Qiao**[2], **Jifeng Dai**[4,2], **Wenhai Wang**[6,2] ✉

[1]Fudan University, [2]Shanghai AI Laboratory,
[3]Shanghai Jiaotong University, [4]Tsinghua University,
[5]Nanjing University, [6]The Chinese University of Hong Kong, [7]SenseTime Research

## Abstract

We construct VisualPRM400K, a dataset comprising about 400K multimodal process supervision data. Building upon this dataset, we develop VisualPRM, an advanced multimodal Process Reward Model (PRM) capable of estimating the value score of each step during the reasoning process. Under the Best-of-N evaluation setting, our model improves the reasoning performance of three types of MLLMs and four different model scales. Even when applied to the highly capable InternVL2.5-78B, it achieves a 5.9-point improvement across seven multimodal reasoning benchmarks. Experimental results show that the PRM model trained on our VisualPRM400K exhibits superior performance compared to Outcome Reward Models and Self-Consistency during BoN evaluation. To further facilitate the development of multimodal PRMs, we construct VisualProcessBench, a benchmark designed to measure the abilities of PRMs and MLLMs to detect incorrect steps in multimodal reasoning tasks. We hope that our work can inspire more future research and contribute to the development of MLLMs. Our model, data, and benchmark will be released.

## 1 Introduction

With the remarkable success of Large Language Models (LLMs) (Touvron et al., 2023a;b; Dubey et al., 2024; Bai et al., 2023a; Team, 2023; Cai et al., 2024; Brown et al., 2020; Achiam et al., 2023; Anthropic, 2024) in natural language processing, Multimodal Large Language Models (MLLMs) (Wang et al., 2023c; Li et al., 2023; Liu et al., 2023a; 2024a; Bai et al., 2023b; Wang et al., 2024e;d; Chen et al., 2023; 2024c;b; OpenAI, 2024; Reid et al., 2024; Yao et al., 2024) have also achieved significant advancements across various vision-language tasks. Despite their strong performance in perception and recognition, a large gap remains in reasoning capabilities between open-source and proprietary models. A series of studies have explored methods to enhance reasoning abilities, focusing on the perspectives of data collection and construction (Muennighoff et al., 2025; Toshniwal et al., 2025; Li et al., 2024b; Liu et al., 2024b), offline preference optimization (Pang et al., 2024; Wang et al., 2024c; Lai et al., 2024), and online reinforcement learning (Shao et al., 2024; Guo et al., 2025; Ahmadian et al., 2024; Hu, 2025). Additionally, another line of research Snell et al. (2024); Dong et al. (2024); Zhang et al. (2025); Wang et al. (2023a) investigates utilizing Test-Time Scaling (TTS) to enhance the reasoning abilities of LLMs. This approach requires the policy model to generate multiple response candidates and select the best one, based on the quality estimation of a critic model, thereby improving the response quality at the cost of higher inference time. However, TTS for MLLMs remains largely unexplored.

This work investigates the application of TTS for MLLMs, focusing on the Best-of-N (BoN) evaluation strategies. The challenges of adapting TTS for MLLMs involves: (1) *Lack of effective critic models.* In BoN evaluation, a critic model is required to estimate the quality of each response candidate. However, as shown in Figure 1, existing open-source MLLMs struggle to serve as critic models, leading to marginal improvements compared to models without TTS. This limitation stems from the lack of sufficient critic data in their training corpus. (2) *Lack of evaluation benchmarks*

*for multimodal critic models.* The effectiveness of TTS heavily depends on the performance of the critic model. However, directly evaluating critics under BoN settings poses two key issues. First, the evaluation cost of BoN is expensive. Although the focus is on the performance of critic models, the policy model is required to generate $N$ reasoning processes, with the majority of computational costs arising from the policy model. Second, BoN performance is also affected by the policy model, making it difficult to compare different critic models when paired with varying policy models.

To solve these challenges, we first introduce VisualPRM400K, a dataset comprising approximately 400K multimodal process supervision data. Each sample includes an image, a question, a step-by-step solution, and correctness annotations for each step. Specifically, we collect question prompts from MMPR v1.1 (Wang et al., 2024c) and then generate process correctness using an automatic data pipeline (Wang et al., 2023a). This pipeline samples multiple continuations starting from a certain step and computes the expected accuracy of that step as the average accuracy of its continuations.

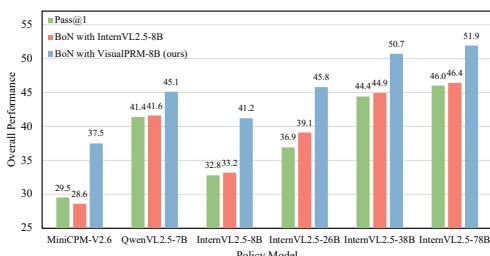

Figure 1: **The overall Best-of-8 evaluation results across seven multimodal reasoning benchmarks with different critic models.** Our VisualPRM greatly enhances the overall performance, while InternVL2.5-8B struggles to be an effective critic model.

To facilitate the evaluation of multimodal critic models, we introduce VisualProcessBench, a benchmark for evaluating PRMs and MLLMs in detecting erroneous steps in multimodal reasoning tasks. This benchmark includes 2,866 samples with 26,950 human-annotated step-wise correctness labels. Each sample includes a multimodal reasoning question, a step-by-step solution, and correctness labels for each step. To ensure annotation accuracy, we employ human experts with at least a university degree to manually assess the correctness of each step. Unlike prior benchmarks (Zheng et al., 2024; Lightman et al., 2023), which require identifying only the first erroneous step, VisualProcessBench challenges models to detect all errors within a given solution. This adjustment aligns with recent advancements in model reflection abilities, helping to reduce false negatives in evaluations. Evaluation results reveal that existing open-source MLLMs struggle to accurately assess step-wise correctness, highlighting the need for improved multimodal critic models.

Building upon the dataset and benchmark, we develop VisualPRM, an advanced multimodal Process Reward Model (PRM) with 8B parameters, to serve as the critic model in BoN evaluation. Each training sample is formulated as a multi-turn chat. The first turn includes the image, the question, and the first solution step, while each subsequent turn presents a new step. The model is trained to predict the correctness of the given step at each turn. *Experimental results demonstrate that VisualPRM enhances MLLM reasoning across different model families and scales.* Specifically, VisualPRM improves the overall reasoning performance of MiniCPM-V2.6, QwenVL2.5-7B, InternVL2.5-8B, and InternVL2.5-78B by 8.0, 3.7, 8.4, and 5.9 points, respectively, across seven multimodal reasoning benchmarks (Yue et al., 2024; Lu et al., 2023; Wang et al., 2024b; Zhang et al., 2024b; Zou et al., 2024; Qiao et al., 2024; Xiao et al., 2024). Additionally, we compare PRMs with Outcome Reward Models and Self-Consistency in BoN evaluation, finding that PRMs consistently outperform both approaches.

In summary, our main contributions are as follows:

(1) We introduce VisualPRM400K, a dataset comprising approximately 400K multimodal process supervision data. Building upon this dataset, we develop VisualPRM, an advanced multimodal PRM to serve as the critic model in the BoN evaluation.

(2) We construct VisualProcessBench, a benchmark designed to measure the abilities of PRMs and MLLMs to identify erroneous steps in multimodal reasoning tasks. This benchmark comprises 2,866 samples with a total of 26,950 human-annotated step-wise correctness labels.

(3) Through extensive experiments, we demonstrate that PRMs can serve as effective critic models for test-time scaling of MLLMs. Specifically, VisualPRM enhances the overall multimodal reasoning performance of MiniCPM-V2.6, QwenVL2.5-7B, InternVL2.5-8B, and InternVL2.5-78B by 8.0, 3.7, 8.4, and 5.9 points, respectively, across seven multimodal reasoning benchmarks. Further-

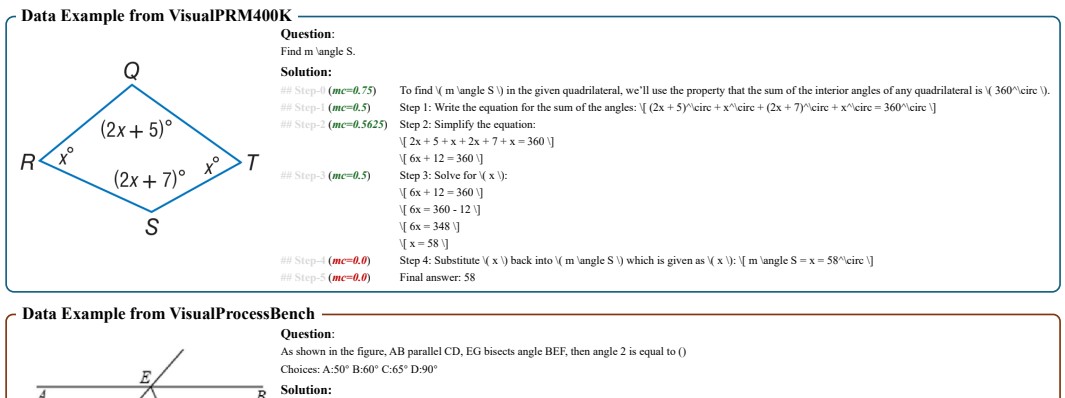

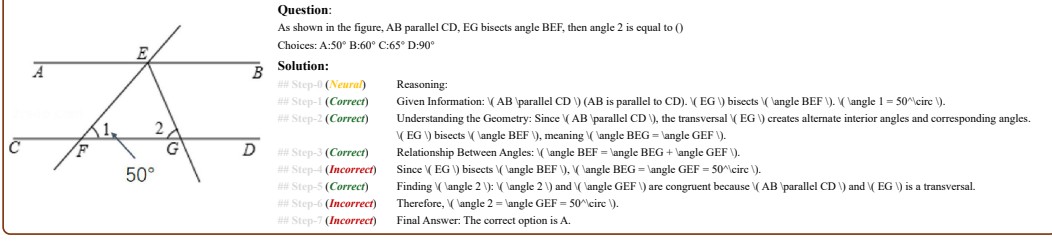

Figure 2: **Data examples in VisualPRM400K and VisualProcessBench.** For VisualPRM400K, we generate the data using an automatic data pipeline. The key idea is to estimate the expected accuracy $mc_i$ of the given step $s_{\leq i}$ based on Monte Carlo sampling and consider the step correct if $mc_i > 0$. During the training process of VisualPRM, the data is formulated as multi-turn conversations and the model is required to predict the correctness of each step conditioned on the image, question, and previous steps. For VisualProcessBench, we collect questions from existing multimodal reasoning benchmarks and generate the solutions using leading MLLMs. Based on these questions and solutions, we employ a team of human experts with at least a university degree to manually annotate the correctness of each step in the solutions.

more, our results show that PRMs consistently outperform both ORMs and SC in BoN evaluation. Additionally, experiments on VisualProcessBench reveal that existing open-source MLLMs struggle to accurately assess the correctness of each step.

## 2 RELATED WORK

**Multimodal Large Language Models.** A wide range of efforts has been made to advance the development of MLLMs, including improvements in model architecture, data construction, and training algorithms. From an architectural perspective, several studies (Liu et al., 2023a; 2024a; Yao et al., 2024; Chen et al., 2024b;c; Wang et al., 2024e;d; Liu et al., 2023b; Wang et al., 2024f; Bai et al., 2025; Yao et al., 2024) employ connectors to align visual embeddings from Vision Foundation Models (VFMs) (Chen et al., 2023; Zhai et al., 2023) with the latent space of LLMs (Bai et al., 2023a; Touvron et al., 2023a;b; Team, 2023), achieving promising performance. Another series of works (Alayrac et al., 2022; Dubey et al., 2024; Tian et al., 2024; Wang et al., 2023b) extends pre-trained LLMs with additional layers to fuse visual features, reducing the number of required visual tokens while introducing extra training cost. In terms of data construction, recent studies have made significant progress (Schuhmann et al., 2022; Zhu et al., 2024; Laurençon et al., 2024; Li et al., 2024b; Liu et al., 2024b; Wang et al., 2024c; Zhao et al., 2025; Dong et al., 2025; Cao et al., 2025). For example, OmniCorpus (Li et al., 2024b) offers a noisy but large-scale multimodal corpus for pre-training, while MMInstruct (Liu et al., 2024b) provides an open-source, high-quality instruction-tuning dataset. Additionally, MMPR (Wang et al., 2024c) constructs a preference dataset focusing on multimodal reasoning abilities. Regarding training algorithms, the InternVL2.5 series (Chen et al., 2024b; Wang et al., 2024c) proposes square loss and Mix Preference Optimization to enhance MLLM capabilities. Despite these advancements, existing works primarily focus on the training process of MLLMs, leaving Test-Time Scaling (TTS) for MLLMs largely underexplored. In this work, we investigate TTS applications for MLLMs, specifically focusing on the Best-of-N evaluation to improve multimodal reasoning performance.

Figure 3: **Different modeling methods for PRMs.** PRMs are developed to estimate the quality of each step in a given solution. For value-based PRMs, the quality of a certain step is determined by its expected accuracy $mc_i$, where a step is considered correct if $mc_i > 0$. For advantage-based PRMs, the quality of a certain step is determined by the improvement of $mc_i$ over $mc_{i-1}$, where a step is considered good if $mc_i - mc_{i-1} > 0$. During the training stage, the output space of PRMs is discretized into specific tokens, while during the inference stage, we compute the step score as the weighted sum of the generation probability for these discretized tokens.

**Process Reward Models.** Reward models play a crucial role in Reinforcement Learning (RL) (Schulman et al., 2017; Shao et al., 2024; Ahmadian et al., 2024; Hu, 2025) and TTS (Snell et al., 2024; Wang et al., 2023a; Dong et al., 2024; Luo et al., 2024). Outcome Reward Models (ORMs) (McAleese et al., 2024; Zhang et al., 2024a; Wang et al., 2024a) directly assign an overall score to the given response. In contrast, Process Reward Models (PRMs) first estimate the quality of each step in the given response and then aggregate them into a final score. PRM800K (Lightman et al., 2023) is the first open-source process supervision dataset, entirely annotated by human experts. To reduce annotation costs, MathShepherd (Wang et al., 2023a) and OmegaPRM (Luo et al., 2024) introduce a Monte Carlo sampling-based data pipeline to automatically estimate the quality of each step. Despite these advancements in natural language processing, multimodal PRMs remain largely underexplored. In this work, we introduce VisualPRM400K, the first multimodal process supervision dataset, and develop VisualPRM, a multimodal PRM trained on this dataset.

**Benchmarks for Reward Models.** The evaluation of reward models (RMs) is a crucial research topic. A series of benchmarks (Lambert et al., 2024; Li et al., 2024a; Liu et al., 2024c) have been proposed to assess the effectiveness of RMs, typically formulated as a binary preference judgment task. Building on this, subsequent work (Zhou et al., 2024) extends the evaluation settings and includes both pairwise and Best-of-N evaluations, providing a more comprehensive evaluation of RM performance. With the rapid advancement of PRMs, a series of benchmarks (Zheng et al., 2024; Song et al., 2025) have been introduced to evaluate their step-wise judgment capabilities. Despite these developments, there remains a lack of a multimodal process benchmark. To bridge this gap and support the development of multimodal PRMs, we introduce VisualProcessBench, a benchmark designed to evaluate the ability of PRMs and MLLMs to detect erroneous steps in multimodal reasoning tasks.

## 3 METHOD

During Best-of-N (BoN) evaluation, a critic model is required to estimate the quality of each response candidate. In this work, we formulate the critic model as a Process Reward Model (PRM). To develop a multimodal PRM, we first construct VisualPRM400K, a dataset comprising about 400K multimodal process supervision data, as detailed in Section 3.1. We then describe our PRM's modeling approach in Section 3.2. Furthermore, to support the evaluation of critic models, we establish VisualProcessBench to measure the abilities of critic models to detect erroneous steps in multimodal reasoning, as introduced in Section 3.3.

### 3.1 VISUALPRM400K

**Definition.** As shown in Figure 2, each data sample in our VisualPRM400K consists of an image $I \in \mathcal{I}$, a question $q \in \mathcal{Q}$, a step-by-step solution $s = \{s_0, s_1, \cdots, s_n\} \in \mathcal{S}$, and the expected accuracy annotation $mc = \{mc_0, mc_1, \cdots, mc_n\}, mc_i \in \mathbb{R}_{\geq 0}$ for each step, where $n$ is the number of steps of a certain solution and $mc_i$ denotes the expected accuracy of step $s_i$. The image sets $\mathcal{I}$ and question sets $\mathcal{Q}$ are collected from MMPR v1.1 (Wang et al., 2024c), while the step-by-step solutions $\mathcal{S}$ are sampled using InternVL2.5 series models (Chen et al., 2024b; Wang et al., 2024c).

**Process Supervision Generation.** Given an image $I$, a question $q$, and a solution $s = \{s_0, s_1, \cdots, s_n\}$, we annotate the correctness of each step $s_i$ using an automatic data pipeline. The key idea is to estimate the expected accuracy of given steps $s_{\leq i}$ based on Monte Carlo sampling. Specifically, the model is required to complete the solution as follows:

$$\tilde{s}_{>i} \sim M(\tilde{s}_{>i} \mid I, q, s_{\leq i}), \tag{1}$$

where $\tilde{s}_{>i}$ is the completion of $s_{\leq i}$. Besides, the expected accuracy of $s_i$ is defined as:

$$mc_i = \frac{\text{num(correct completions)}}{\text{num(sampled completions)}}. \tag{2}$$

Notably, to reduce the data construction costs, we set the max number of steps to 12 and evenly merge the steps if the number of current steps exceeds the threshold.

**Statistics.** During the construction process, we sample 4 solutions for each image-question pair and split each of them into at most 12 steps. For each step, we sample 16 continuations and compute $m_i$ according to these continuations. The resulting dataset comprises approximately 400K samples and 2 million steps with process supervision. Each response averages 126.9 words and 5.6 steps, while each step averages 22.6 words. Among these steps, about 10% are incorrect steps. Despite the imbalanced distribution of correct and incorrect steps, our PRM demonstrates promising performance, as shown in Section 4.

## 3.2 VISUALPRM

**Overview.** During the training process, we formulate the process supervision problem as a multi-turn chat task so that we can effectively leverage the generation ability of MLLMs. The image $I$, question $q$, and the first step $s_0$ of the solution to this question are included in the first turn and a new step is presented in each subsequent turn. The model is required to predict the quality of the given step in each turn as follows:

$$y_i \sim M(y_i \mid I, q, s_{\leq i}), \tag{3}$$

where $y_i$ denotes the quality of $i$-th step.

**For value-based PRMs,** the quality of a certain step is determined by its expected accuracy $mc_i$, which is similar to the definition of the value function in reinforcement learning. Following Math-Shepherd (Wang et al., 2023a; Dong et al., 2024), we require the model to predict the correctness $c_i \in \{+, -\}$ of the given step, rather than the exact score of $mc_i$. The $i$-th step is considered correct if $mc_i > 0$. We also try to set a threshold to reduce false positive steps, but find that such a threshold negatively impacts the PRM performance, as shown in Section B. Notably, unlike previous works (Wang et al., 2023a; Lightman et al., 2023; Dong et al., 2024), which choose to supervise only up to the first incorrect step, we always supervise all steps.

**For advantage-based PRMs,** the quality of a certain step is determined by the improvement of $mc_i$ over $mc_{i-1}$, which is analogous to the definition of the advantage function in reinforcement learning. Similar to value-based PRMs, the quality space is discretized into predefined values $\{+, =, -\}$, meaning that the $i$-th step $s_i$ results in a superior, comparable, or inferior situation.

**During inference stage,** we first compute the scores of each step and then merge them to obtain the response score. Specifically, the score for each step is defined as the weighted sum of the generation probability for the discretized scores. For value-based PRMs, the weights for $\{+, -\}$ are $\{1, 0\}$. For advantage-based PRMs, the weights for $\{+, =, -\}$ are $\{1, 0, -1\}$. Without further explanation, we average the scores of each step as the response score.

## 3.3 VISUALPROCESSBENCH

**Definition.** Each sample in our benchmark consists of a multimodal reasoning question, a step-by-step solution, and correctness annotations for each step. Considering that recent models begin to demonstrate reflection abilities to rectify their own reasoning process, the evaluation setting used in previous works (Zheng et al., 2024; Lightman et al., 2023), which only requires the model to find the first erroneous step, may lead to a false negative estimation. Therefore, our benchmark requires the model to identify all erroneous steps in the given solution instead of only the first erroneous step.

**Data Source.** Our benchmark focuses on multimodal reasoning tasks, collecting images and questions from existing representative multimodal reasoning benchmarks, including MMMU (Yue et al., 2024), MathVision (Wang et al., 2024b), MathVerse (Zhang et al., 2024b), DynaMath (Zou et al., 2024), and WeMath (Qiao et al., 2024). Given these questions, we generate step-by-step solutions using leading MLLMs, including GPT-4o (OpenAI, 2024), Claude-3.5-Sonnet (Anthropic, 2024), Gemini-2.0-Flash (Team et al., 2023), QvQ-72B-Preview (Team, 2024), and InternVL2.5-78B (Chen et al., 2024b). The solutions are sampled from different MLLMs to ensure their diversity.

**Step Correctness Annotation.** We employ a team of human experts with at least a university degree to manually annotate the correctness of each step in the solutions. Specifically, 13 people worked for 3 days, resulting in a workload of 39 person-days. The cost per person-day is approximately 37 dollars. During the annotation process, annotators are provided with the image, question, ground truth answer, and each step of the solution. Their task is to assign each step in the solution a label of positive, negative, or neutral, as illustrated in Figure 2. A positive label indicates that the step is correct, while a negative label signifies an incorrect step. The neural label is assigned to steps that do not involve any reasoning process or provide no additional information. To ensure the annotation quality, annotators are permitted to skip questions they do not understand. During the annotation process, our dataset is divided into 10 splits, each containing approximately 300 samples. For each split, the authors of this paper manually review about 10% of the samples. Splits with erroneous annotations are sent back for re-annotation. See Section F for more data examples.

Table 1: **Statistics of VisualProcessBench.**

| Statistics | Number |
|---|---|
| Total Samples | 2866 |
| - MMMU | 267 |
| - MathVision | 712 |
| - MathVerse | 1026 |
| - DynaMath | 570 |
| - WeMath | 291 |
| Source Solutions | 2866 |
| - GPT-4o | 870 |
| - Claude-3.5-Sonnet | 865 |
| - QvQ-72B-Preview | 825 |
| - InternVL2.5-78B | 306 |
| Total Steps | 26950 |
| - Correct Steps | 16585 |
| - Incorrect Steps | 7691 |
| - Neural Steps | 2674 |
| Query Word Length Quartile | (22, 24, 50) |
| Response Word Length Quartile | (137, 193, 552) |
| Step Word Length Quartile | (13, 31, 67) |
| Number of Steps per Solution | 9.4 |

**Statistics.** As shown in Table 1, our benchmark comprises 2866 samples. To enhance the diversity of our evaluation samples, we gather questions and solutions from a wide range of benchmarks and models while carefully regulating their distribution. The statistics of step distribution are presented in Section D.

**Metrics.** In this work, we use macro F1 scores to compare model performance, aiming to mitigate the impact of the imbalanced distribution between correct and incorrect steps. Specifically, we first compute the F1 scores separately for correct and incorrect steps and then take their average to obtain the overall score.

## 4 EXPERIMENTS

In this section, we first employ VisualPRM to evaluate various MLLMs using BoN evaluation strategies in Section 4.1, demonstrating that PRMs can significantly enhance the reasoning abilities of MLLMs. Next, we evaluate our VisualPRM and other leading MLLMs on VisualProcessBench in Section 4.2. Finally, the ablation studies are presented in Section 4.3 and Section B.

### 4.1 RESULTS WITH BEST-OF-N EVALUATION

**Benchmarks.** We evaluate the reasoning abilities of MLLMs across seven benchmarks, including MMMU (Yue et al., 2024), MathVista (Lu et al., 2023), MathVision (Wang et al., 2024b), MathVerse (Zhang et al., 2024b), DynaMath (Zou et al., 2024), WeMath (Qiao et al., 2024), and LogicVista (Xiao et al., 2024). The evaluation samples include subject-based, mathematical, and logical reasoning problems. We report the worst-case accuracy for DynaMath and the overall accuracy for the remaining benchmarks. For MathVerse, we report the performance on the Vision-Only split.

**Settings.** Without further explanation, we use VisualPRM as the critic model for BoN evaluation and set $N$ to 8 by default. The policy model is required to generate $N$ distinct step-by-step Chain-of-Thought (CoT) reasoning processes with a temperature of 0.7. The response with the highest score is then selected to determine the correctness.

Table 2: **Results on seven multimodal reasoning benchmarks.** MMMU is a multidisciplinary reasoning benchmark. MathVista, MathVision, MathVerse, DynaMath, and WeMath are mathematics benchmarks. For MathVerse, we report the performance on Vision-Only (VO) split. LogicVista is a logical reasoning benchmark. Part of the results are collected from the OpenCompass leaderboard. The overall score is the average score of the above benchmarks.

| Model | MMMU | MathVista | MathVision | MathVerse-VO | DynaMath | WeMath | LogicVista | Overall |
|---|---|---|---|---|---|---|---|---|
| *Proprietary Models* | | | | | | | | |
| GPT-4o | 70.7 | 60.0 | 31.2 | 40.6 | 34.5 | 45.8 | 52.8 | 47.9 |
| Gemini-2.0-Flash | 69.9 | 70.4 | 43.6 | 47.8 | 42.1 | 47.4 | 52.3 | 53.4 |
| Claude-3.5-Sonnet | 66.4 | 65.3 | 35.6 | 46.3 | 35.7 | 44.0 | 60.4 | 50.5 |
| *Open-source Models* | | | | | | | | |
| MiniCPM-V2.6-8B | 49.8 | 60.8 | 23.4 | 18.9 | 9.8 | 16.4 | 27.5 | 29.5 |
| +VisualPRM | 56.8 | 65.7 | 24.7 | 35.8 | 11.2 | 31.0 | 37.4 | 37.5 |
| | **+7.0** | **+4.9** | **+1.3** | **+16.9** | **+1.4** | **+14.6** | **+9.8** | **+8.0** |
| Qwen2.5-VL-7B | 55.0 | 67.8 | 25.4 | 41.1 | 21.0 | 35.2 | 44.1 | 41.4 |
| +VisualPRM | 58.6 | 70.3 | 31.3 | 44.3 | 23.0 | 39.8 | 48.3 | 45.1 |
| | **+3.6** | **+2.5** | **+5.9** | **+3.2** | **+2.0** | **+4.6** | **+4.2** | **+3.7** |
| InternVL2.5-8B | 56.2 | 64.5 | 17.0 | 22.8 | 9.4 | 23.5 | 36.0 | 32.8 |
| +VisualPRM | 60.2 | 68.5 | 25.7 | 35.8 | 18.0 | 36.5 | 43.8 | 41.2 |
| | **+4.0** | **+4.0** | **+8.7** | **+13.0** | **+8.6** | **+13.0** | **+7.8** | **+8.4** |
| InternVL2.5-26B | 60.7 | 68.2 | 23.4 | 24.0 | 11.4 | 30.9 | 39.6 | 36.9 |
| +VisualPRM | 63.9 | 73.1 | 29.6 | 39.1 | 23.2 | 40.8 | 51.0 | 45.8 |
| | **+3.2** | **+4.9** | **+6.2** | **+15.1** | **+11.8** | **+9.9** | **+11.4** | **+8.9** |
| InternVL2.5-38B | 63.9 | 71.9 | 32.2 | 36.9 | 20.0 | 38.3 | 47.9 | 44.4 |
| +VisualPRM | 69.0 | 73.9 | 35.2 | 46.7 | 30.5 | 46.2 | 53.7 | 50.7 |
| | **+5.1** | **+2.0** | **+3.0** | **+9.8** | **+10.5** | **+7.9** | **+5.8** | **+6.3** |
| InternVL2.5-78B | 70.0 | 72.3 | 32.2 | 39.2 | 19.2 | 39.8 | 49.0 | 46.0 |
| +VisualPRM | 70.7 | 75.1 | 35.9 | 47.1 | 31.3 | 49.1 | 53.9 | 51.9 |
| | **+0.7** | **+2.8** | **+3.7** | **+7.9** | **+12.1** | **+9.3** | **+4.9** | **+5.9** |

**Results.** As shown in Table 2, *VisualPRM greatly enhances the reasoning abilities of MLLMs across different model scales and families.* Specifically, for models with fewer than 10 billion parameters, the overall performance of InternVL2.5-8B, MiniCPM-V-8B, and Qwen2.5-VL-7B improves by 8.4, 8.0, and 3.7 points, respectively, demonstrating the effectiveness of test-time scaling across different model families. For larger models, InternVL2.5-26B, InternVL2.5-38B, and InternVL2.5-78B also achieve substantial performance gains over their counterparts without TTS, further validating the scalability and effectiveness of TTS across different model sizes.

## 4.2 RESULTS ON VISUALPROCESSBENCH

**Settings.** For the evaluation of PRMs, a step is considered correct if the probability of outputting "+" exceeds that of outputting "−" by a certain threshold. For the evaluation of MLLMs, the model is prompted to analyze each step and determine its correctness, classifying it as either correct or incorrect. When computing the F1 score, we exclude steps labeled as neural by human annotators in Section 3.3.

**Results.** As shown in Table 3, most existing MLLMs struggle to accurately assess the correctness of each step. Specifically, the overall F1 score for random guessing is 50.0, while most open-source MLLMs achieve scores close to this baseline, highlighting their limitations as critic models. We manually check the judgments of these open-source MLLMs and observe that these models tend to provide positive analysis and label most steps as correct. For example, InternVL2.5-8B achieves an F1 score of 76.8 for positive steps, while its F1 score for negative steps is only 19.2, indicating that InternVL2.5-8B rarely identifies steps as incorrect. Furthermore, compared to proprietary models, our VisualPRM demonstrates competitive performance, achieving an overall F1 score of 62.0—outperforming GPT-4o and GPT-4o-Mini, and performing on par with Gemini-2.0-Flash. Notably, our model, with only 8 billion parameters, is more efficient than these proprietary counterparts.

## 4.3 ABLATION STUDIES

**Effects of BoN.** The training data for ORM are nearly identical to those used for PRM,

Table 3: **Results on VisualProcessBench.** We report the macro F1 of the correct and incorrect steps. The overall score is the micro average of the score from different data sources.

| Model | MMMU | MathVision | MathVerse-VO | DynaMath | WeMath | Overall |
|---|---|---|---|---|---|---|
| Random Guessing | 50.0 | 50.0 | 50.0 | 50.0 | 50.0 | 50.0 |
| *Proprietary Models* | | | | | | |
| GPT-4o-Mini | 53.6 | 58.9 | 57.1 | 56.7 | 58.5 | 57.9 |
| GPT-4o | 56.3 | 60.2 | 59.7 | 59.0 | 63.3 | 60.3 |
| Gemini-2.0-Flash | 58.5 | 60.1 | 62.8 | 66.7 | 58.7 | 62.3 |
| *Open-source Models* | | | | | | |
| MiniCPM-V2.6-8B | 44.9 | 50.9 | 58.9 | 46.7 | 57.4 | 50.4 |
| LLaVA-OV-7B | 45.7 | 43.0 | 42.2 | 44.7 | 52.5 | 44.4 |
| LLaVA-OV-72B | 46.1 | 48.4 | 53.0 | 57.0 | 57.3 | 52.3 |
| Qwen2.5-VL-7B | 53.1 | 51.8 | 47.8 | 51.3 | 54.2 | 51.0 |
| Qwen2.5-VL-72B | 59.2 | 59.0 | 59.7 | 62.9 | 62.3 | 60.5 |
| InternVL2.5-8B | 47.1 | 45.5 | 47.8 | 50.3 | 50.8 | 48.0 |
| InternVL2.5-26B | 48.8 | 47.4 | 49.2 | 50.4 | 51.4 | 49.2 |
| InternVL2.5-38B | 51.5 | 48.4 | 50.9 | 51.8 | 52.5 | 50.8 |
| InternVL2.5-78B | 52.0 | 51.7 | 53.7 | 50.8 | 52.5 | 52.6 |
| VisualPRM (ours) | 58.5 | 62.1 | 61.0 | 62.7 | 61.8 | 62.0 |

except that all steps are concatenated into a single step, and step-wise correctness annotations are converted into a single correctness label for the outcome. Here, we increase the number of response candidates sampled from InternVL2.5-8B and select the final response using Self-Consistency (SC) (Wang et al., 2022), Outcome Reward Model (ORM), and PRM. As shown in Figure 4, increasing the number of response candidates $N$ improves the reasoning performance of InternVL2.5-8B and MiniCPM-V2.6-8B when using SC, ORM, or PRM, with PRM yielding the most significant improvements. Specifically, when using InternVL2.5-8B as the policy model, PRM outperforms SC and ORM

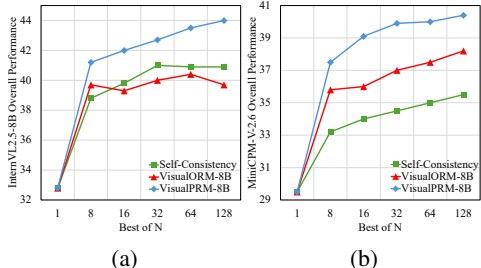

(a)         (b)

Figure 4: **Overall Best-of-N results across seven multimodal reasoning benchmarks with different policy and critic models.**

by 2.4 and 1.5 points, respectively, under the Best-of-8 evaluation setting. Moreover, this performance gap widens as $N$ increases, reaching 3.1 and 4.3 points when $N$ is set to 128. Notably, when using ORM as the critic model, although performance improves during Best-of-8 evaluation, further increasing $N$ does not lead to consistent gains for InternVL2.5-8B. For example, the Best-of-128 performance is inferior to the Best-of-64 performance.

**Effects of PRM modeling methods.** Here, we compare the value-based PRM and the advantage-based PRM introduced in Section 3.2, along with different methods for aggregating step scores into a final score, including averaging, as well as selecting the maximum or minimum value. The results are presented in Table 4. We find that value-based PRMs outperform advantage-based PRMs in both BoN evaluation settings and VL-ProcessBench. We attribute this to the inherent noise in our training data, which is generated through an automatic data pipeline, making it challenging to accurately determine whether a given step contributes to higher or lower expected accuracy. We also compare two training strategies: supervising all steps (*i.e.*, w/o early stop) versus supervising only up to the first incorrect step (*i.e.*, w. early stop) during training. Experimental results show that the former yields better performance. Regarding different score aggregation methods, we find that selecting the maximum value results in poorer performance compared to averaging

Table 4: **Comparison of different critic models and score aggregation methods.**

| Critic Model | BoN | VL-ProcessBench |
|---|---|---|
| Pass@1 | 32.8 | - |
| Random Guessing | 33.0 | 50.0 |
| InternVL2.5-8B | 33.2 | 48.0 |
| InternVL2.5-78B | 34.2 | 52.6 |
| Advantage-based PRM | | |
| +Min | 36.8 | 55.0 |
| +Max | 36.9 | 55.0 |
| +Average | 37.4 | 55.0 |
| Value (w. early stop) | | |
| +Min | 40.3 | 61.6 |
| +Max | 37.0 | 61.6 |
| +Average | 40.6 | 61.6 |
| Value (w/o early stop) | | |
| +Min | 40.4 | 62.0 |
| +Max | 35.9 | 62.0 |
| +Average | 41.1 | 62.0 |

or taking the minimum value. Analyzing the generated scores reveals that most responses contain a high-scored step, close to 1, at the beginning of the solution. This phenomenon likely arises because

most erroneous steps appear in the middle of the solution. Our statistics of VisualProcessBench presented in Section D further demonstrate this conclusion. Furthermore, averaging performs better than selecting the maximum value, likely because the latter relies on a single step's score, while averaging accounts for multiple steps and can be considered as an ensemble approach, which benefits the step quality estimation.

**MLLM-as-a-Judger.** Existing MLLMs can be prompted to serve as a critic model. However, as shown in Table 4, the InternVL2.5 series struggle to improve BoN performance, resulting in only marginal improvements. Upon analyzing the generated scores, we find that these models assign similar scores to most solutions. Consistent with our observations in Section 4.2, the InternVL2.5 series tend to generate positive judgments for most steps, which hinders their ability to effectively distinguish and select the truly superior response. In addition to their effectiveness as critic models for MLLMs, their inference latency also limits efficiency. Specifically, MLLMs generate judgments for each step in an autoregressive manner, which is time-consuming. In contrast, our VisualPRM computes scores for all steps in a single forward pass by using a "+" as a placeholder for model responses and interpreting its generation probability as the step score.

**Results on text-only performance.** To assess the effectiveness of VisualPRM on text-only inputs, we evaluate the Qwen2.5 series (Yang et al., 2024) and InternVL2.5 series (Chen et al., 2024b) on three text reasoning benchmarks under BoN evaluation settings: GSM8K (Cobbe et al., 2021), MATH-500 (Hendrycks et al., 2021), and GPQA-Diamond (Rein et al., 2024). We report accuracy as the evaluation metric for these benchmarks. As shown in Table 5, our model enhances the text reasoning abilities of both the Qwen2.5 series and the InternVL2.5 series. Specifically, Qwen2.5-7B achieves improvements of 6.1 and 5.0 points on MATH-500 and GPQA-Diamond, respectively. Similarly, Qwen2.5-72B demonstrates gains of 2.1 and 6.6 points on these benchmarks. For the InternVL2.5 series, InternVL2.5-8B, InternVL2.5-38B, and InternVL2.5-78B achieve improvements of 9.4 and 5.0, 4.6 and 8.1, and 7.4 and 3.5 points, respectively, on MATH-500 and GPQA-Diamond. These results demonstrate the effectiveness of our VisualPRM in text-only scenarios.

Table 5: **Results on text benchmarks.**

| Model | GSM8K | MATH-500 | GPQA |
|---|---|---|---|
| *Large Language Models* | | | |
| Qwen2.5-7B | 91.6 | 75.5 | 36.4 |
| +VisualPRM | 94.5 | 81.6 | 41.4 |
| | **+2.9** | **+6.1** | **+5.0** |
| Qwen2.5-32B | 95.9 | 83.1 | 49.5 |
| +VisualPRM | 96.1 | 85.4 | 53.5 |
| | **+0.2** | **+2.3** | **+4.0** |
| Qwen2.5-72B | 95.8 | 83.1 | 49.0 |
| +VisualPRM | 96.5 | 85.2 | 55.6 |
| | **+0.7** | **+2.1** | **+6.6** |
| *Multimodal Large Language Models* | | | |
| InternVL2.5-8B | 81.9 | 56.8 | 29.3 |
| +VisualPRM | 82.9 | 66.2 | 34.3 |
| | **+1.1** | **+9.4** | **+5.0** |
| InternVL2.5-38B | 94.6 | 75.4 | 44.4 |
| +VisualPRM | 95.6 | 80.0 | 52.5 |
| | **+1.0** | **+4.6** | **+8.1** |
| InternVL2.5-78B | 93.6 | 70.4 | 47.5 |
| +VisualPRM | 94.5 | 77.8 | 51.0 |
| | **+0.9** | **+7.4** | **+3.5** |

## 5 CONCLUSION

In this work, we construct VisualPRM400K, a dataset comprising about 400K multimodal process supervision data. Building upon this dataset, we develop VisualPRM, an advanced multimodal Process Reward Model (PRM) capable of estimating the value score of each step during the reasoning process. With the Best-of-N (BoN) evaluation strategies, our model improves the reasoning abilities of existing Multimodal Large Language Models (MLLMs) across different model scales and families. Experimental results show that our model exhibits superior performance compared to Outcome Reward Models and Self-Consistency during BoN evaluation. To further facilitate the development of multimodal critic models, we construct VisualProcessBench, a benchmark designed to measure the abilities of PRMs and MLLMs to detect incorrect steps in multimodal reasoning tasks. Evaluation results show that existing open-source MLLMs perform badly on this task. We hope that our work can inspire more research and contribute to the development of MLLMs.

**Limitations.** The main contributions of this study lie in the construction of the training dataset VisualPRM400K and the benchmark VisualProcessBench. Although we introduce VisualPRM as a model baseline, our exploration of training and modeling strategies for multimodal PRMs is limited. We believe that further investigation into these strategies can advance the development of multimodal PRMs, while we also emphasize that our proposed training dataset and benchmark provide a valuable foundation for future research on multimodal PRMs.

ETHICS STATEMENT.

This study adheres to the outlined ICLR ethical guidelines and follows the principles of responsible research. Our dataset and benchmark are constructed from publicly available datasets, which mainly focus on multimodal reasoning tasks. We confirm that no personally identifiable, sensitive, or harmful data are utilized. Although human annotations are required during benchmark construction, they are solely used to assess the correctness of each reasoning step and thus are unrelated to any ethical concerns. Furthermore, we have carefully considered the potential societal implications of our work, including possible misuse, and conclude that our contributions primarily serve to advance scientific knowledge without introducing any foreseeable risks.

REPRODUCIBILITY STATEMENT.

We adhere to the reproducibility guidelines in the ICLR 2026 author guidelines. We will open-source our code, scripts, and data for training and evaluation to ensure the reproducibility of our experiments. Additionally, the training data construction pipeline will be open-sourced to ensure the reproducibility of data construction. We also provide the key hyper-parameters in Appendix A.

ACKNOWLEDGMENTS

The work is supported by the National Key R&D Program of China (NO. 2022ZD0161300), the Youth PhD Student Research Project under the National Natural Science Foundation (No. 623B2050), and the National Natural Science Foundation of China (Grant No. 62376134).

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

Table 6: **Performance after Reinforcement Learning.** The model trained with the additional process reward introduced by VisualPRM achieves better performance than the vanilla GRPO.

| Model | MMMU | MathVista | MathVision | MathVerse | DynaMath | WeMath | LogicVista | Overall |
|---|---|---|---|---|---|---|---|---|
| InternVL2.5-8B | 56.2 | 64.5 | 17.0 | 22.8 | 9.4 | 23.5 | 36.0 | 32.8 |
| +GRPO | 55.9 | 67.2 | 25.7 | 28.0 | 13.6 | 27.5 | 39.1 | 36.7 |
| +Process Reward | 59.0 | 69.2 | 27.0 | 32.1 | 16.8 | 29.1 | 42.7 | 39.4 |

## A  TRAINING HYPER-PARAMETERS

During the training process of VisualPRM, the data-packing strategy (Chen et al., 2024b) is enabled during training. We employ the AdamW optimizer (Loshchilov & Hutter, 2017) with the $\beta_1$ of 0.9, the $\beta_2$ of 0.999, and the weight decay of 0.05. The learning rate is initialized as $1e$-5. The training phases include a linear warmup that lasts until the first 5% of training steps. The warmup is followed by a cosine decay strategy with a minimum learning rate of 0. We set the training epoch to 1.

## B  MORE ABLATION STUDIES

### B.1  EFFECTIVENESS OF VISUALPRM ON REINFORCEMENT LEARNING

In this section, we fine-tune InternVL2.5-8B using GRPO (Shao et al., 2024), a widely used reinforcement learning (RL) algorithm that relies on correctness-based outcome rewards. Building upon GRPO, we integrate VisualPRM to provide additional process-level rewards. Specifically, for each correct response, individual reasoning steps are further rewarded according to the step scores predicted by VisualPRM. The training dataset consists of approximately 30K multimodal questions, randomly downsampled from MMPR v1.1 (Wang et al., 2024c). During training, we set the learning rate to $1e$-6. Each training iteration processes 512 queries, and the model parameters are updated with eight PPO steps (*i.e.*, 32 queries per step). For each query, we sample 16 responses. For evaluation, we use the same benchmarks discussed in Section 4.1. The experimental results, summarized in Table 6, show that GRPO significantly enhances the reasoning capabilities of InternVL2.5-8B, improving its overall performance from 32.8 to 36.7. Furthermore, by incorporating the process reward introduced by our proposed VisualPRM, the overall performance is further boosted to 39.4, demonstrating the effectiveness of the process reward provided by VisualPRM.

### B.2  EFFECTS OF TRAINING HYPER-PARAMETERS

When training our value-based Process Reward Model (PRM) using VisualPRM400K, we define a step as correct if its expected accuracy exceeds 0. In this section, we analyze the impact of varying expected accuracy thresholds for determining step correctness. As shown in Table 7, increasing the threshold results in a decline in both Best-of-8 evaluation performance and VisualProcessBench scores. These results are consistent with the observation in Qwen2.5-Math-PRM (Zhang et al., 2025). Therefore, we suggest setting the threshold to 0 during training.

### B.3  EFFECTS OF GENERATION HYPER-PARAMETERS

In this section, we analyze the impact of generation temperature on the Best-of-8 evaluation. As shown in Table 7, as the temperature increases from 0.3 to 1.3, the overall performance of InternVL2.5-8B first improves and then declines. We attribute this phenomenon to the trade-off between response diversity and accuracy. When the temperature is low (*e.g.*, set to 0.3), the generated responses lack diversity, limiting the model's performance upper bound. Conversely, when the temperature is high (*e.g.*, set to 1.3), the responses become more random, reducing the accuracy of individual responses and lowering the model's overall performance ceiling.

### B.4  EFFECTS OF BEST-OF-N EVALUATION

In this section, we present the Best-of-N evaluation results as $N$ increases, as shown in Table 8 and Table 9. Our results indicate that as $N$ increases, VisualPRM consistently enhances the reasoning

Table 7: **Ablation studies about the effects of expected accuracy threshold and generationo temperaure.**

| Model | MMMU | MathVista | MathVision | MathVerse-VO | DynaMath | WeMath | LogicVista | Overall | VisualProcessBench |
|---|---|---|---|---|---|---|---|---|---|
| *Threshold* | | | | | | | | | |
| Threshold=0.00 | 59.3 | 68.5 | 25.7 | 35.8 | 18.0 | 36.5 | 43.8 | 41.1 | 62.0 |
| Threshold=0.625 | 59.7 | 66.8 | 24.7 | 36.7 | 18.4 | 35.0 | 41.8 | 40.4 | 61.0 |
| Threshold=0.125 | 58.0 | 67.9 | 27.6 | 35.4 | 17.4 | 35.3 | 41.6 | 40.5 | 60.7 |
| Threshold=0.25 | 58.6 | 67.6 | 25.7 | 33.6 | 16.8 | 36.0 | 41.4 | 40.0 | 60.2 |
| *Temperature* | | | | | | | | | |
| Temperature=0.3 | 59.7 | 69.4 | 26.0 | 32.6 | 17.6 | 35.5 | 42.7 | 40.5 | - |
| Temperature=0.7 | 59.3 | 68.5 | 25.7 | 35.8 | 18.0 | 36.5 | 43.8 | 41.1 | - |
| Temperature=1.0 | 61.7 | 67.2 | 27.3 | 35.8 | 16.6 | 34.2 | 43.2 | 40.9 | - |
| Temperature=1.3 | 57.9 | 66.1 | 25.0 | 32.1 | 16.8 | 31.9 | 40.5 | 38.6 | - |

Table 8: **Overall Best-of-N results of InternVL2.5-8B across seven multimodal reasoning benchmarks with different critic models.**

| Model | BoN | MMMU | MathVista | MathVision | MathVerse-VO | DynaMath | WeMath | LogicVista | Overall |
|---|---|---|---|---|---|---|---|---|---|
| Self Consistency | 1 | 56.2 | 64.5 | 17.0 | 22.8 | 9.4 | 23.5 | 36.0 | 32.8 |
| | 8 | 58.0 | 65.9 | 23.4 | 30.5 | 18.4 | 32.7 | 43.0 | 38.8 |
| | 16 | 58.6 | 65.8 | 26.3 | 32.1 | 19.4 | 33.0 | 43.4 | 39.8 |
| | 32 | 60.4 | 66.7 | 28.0 | 32.6 | 20.8 | 34.1 | 44.7 | 41.0 |
| | 64 | 59.7 | 66.7 | 26.6 | 33.2 | 20.6 | 35.8 | 43.4 | 40.9 |
| | 128 | 60.6 | 67.4 | 25.7 | 32.0 | 22.6 | 34.7 | 43.2 | 40.9 |
| VisualORM | 1 | 56.2 | 64.5 | 17.0 | 22.8 | 9.4 | 23.5 | 36.0 | 32.8 |
| | 8 | 60.2 | 67.0 | 25.3 | 32.5 | 16.4 | 35.0 | 41.8 | 39.7 |
| | 16 | 58.3 | 67.7 | 27.0 | 33.6 | 16.6 | 33.1 | 39.1 | 39.3 |
| | 32 | 58.6 | 67.9 | 26.3 | 33.6 | 17.4 | 34.4 | 42.1 | 40.0 |
| | 64 | 59.4 | 66.8 | 28.6 | 33.9 | 17.8 | 34.1 | 42.3 | 40.4 |
| | 128 | 59.4 | 66.6 | 28.3 | 33.5 | 16.8 | 32.3 | 40.9 | 39.7 |
| VisualPRM | 1 | 56.2 | 64.5 | 17.0 | 22.8 | 9.4 | 23.5 | 36.0 | 32.8 |
| | 8 | 60.2 | 68.5 | 25.7 | 35.8 | 18.0 | 36.5 | 43.8 | 41.2 |
| | 16 | 60.2 | 69.9 | 27.3 | 36.4 | 19.0 | 38.8 | 42.5 | 42.0 |
| | 32 | 60.3 | 70.4 | 29.6 | 37.8 | 17.2 | 40.3 | 43.4 | 42.7 |
| | 64 | 61.4 | 69.6 | 30.6 | 38.2 | 18.8 | 40.2 | 45.4 | 43.5 |
| | 128 | 61.7 | 70.8 | 30.3 | 39.3 | 19.4 | 40.9 | 45.4 | 44.0 |

abilities of InternVL2.5-8B (Chen et al., 2024b) and MiniCPM-V2.6 (Yao et al., 2024). Specifically, as $N$ increases from 8 to 128, the overall performance of InternVL2.5-8B improves from 41.2 to 44.0, while MiniCPM-V2.6 improves from 37.5 to 40.4, demonstrating the scalability of Test-Time Scaling for MLLMs.

### B.5 EFFECTS OF ROLLOUTS FROM DIFFERENT MODELS

Here we provide additional analysis of the data pipeline. Specifically, we incorporate additional PRM training data constructed from rollouts sampled using Qwen2.5-VL-7B and MiMo-VL-7B, and we report the BoN performance of PRMs trained on this expanded dataset. We construct about 74K additional samples (44K from Qwen2.5-VL and 30K from MiMo-VL-7B). The construction process is identical to that of VisualPRM400K, with the only difference being the policy model used for sampling. As shown in Table 10, experimental results confirm that incorporating more diverse rollouts from additional models further improves VisualPRM performance. Notably, "+44K rollouts from Qwen2.5-VL" indicates that the training dataset is constructed by adding an additional 44K samples on top of VisualPRM400K.

### B.6 EFFECTS OF MAXIMUM NUMBER OF STEPS

Here, we analyze how the maximum number of steps in data construction influences model performance. To reduce annotation cost, we reused the annotations from the max-step = 12 configuration to build datasets with max-step = 1/3/6/9. We then trained separate VisualPRM models on these subsets. As shown in Table 11, increasing the maximum step count does improve performance, but the gains exhibit clear diminishing returns—for instance, increasing the max steps from 9 to 12 yields only very marginal improvement.

Table 9: **Overall Best-of-N results of MiniCPM-V2.6 across seven multimodal reasoning benchmarks with different critic models.**

| Model | BoN | MMMU | MathVista | MathVision | MathVerse-VO | DynaMath | WeMath | LogicVista | Overall |
|---|---|---|---|---|---|---|---|---|---|
| Self Consistency | 1 | 49.8 | 60.8 | 23.4 | 18.9 | 9.8 | 16.4 | 27.5 | 29.5 |
| | 8 | 51.8 | 58.9 | 21.7 | 31.5 | 10.0 | 22.6 | 35.6 | 33.2 |
| | 16 | 51.7 | 60.2 | 21.7 | 31.5 | 11.6 | 25.7 | 35.3 | 34.0 |
| | 32 | 52.2 | 60.1 | 24.3 | 33.1 | 11.4 | 24.3 | 36.0 | 34.5 |
| | 64 | 51.7 | 61.0 | 23.4 | 34.8 | 12.8 | 25.8 | 35.3 | 35.0 |
| | 128 | 53.2 | 61.7 | 25.7 | 33.5 | 13.0 | 25.6 | 35.6 | 35.5 |
| VisualORM | 1 | 49.8 | 60.8 | 23.4 | 18.9 | 9.8 | 16.4 | 27.5 | 29.5 |
| | 8 | 55.7 | 66.0 | 22.0 | 33.5 | 10.2 | 24.1 | 38.9 | 35.8 |
| | 16 | 56.4 | 65.3 | 24.0 | 32.1 | 10.4 | 27.3 | 36.5 | 36.0 |
| | 32 | 58.8 | 64.8 | 19.7 | 35.7 | 12.0 | 29.4 | 38.5 | 37.0 |
| | 64 | 58.2 | 67.3 | 22.7 | 35.5 | 11.0 | 30.1 | 37.6 | 37.5 |
| | 128 | 58.2 | 66.5 | 25.3 | 35.4 | 11.6 | 30.0 | 40.7 | 38.2 |
| VisualPRM | 1 | 49.8 | 60.8 | 23.4 | 18.9 | 9.8 | 16.4 | 27.5 | 29.5 |
| | 8 | 56.8 | 65.7 | 24.7 | 35.8 | 11.2 | 31.0 | 37.4 | 37.5 |
| | 16 | 58.8 | 68.6 | 24.0 | 37.3 | 12.4 | 32.7 | 39.8 | 39.1 |
| | 32 | 57.8 | 68.4 | 26.6 | 38.5 | 13.4 | 35.3 | 39.1 | 39.9 |
| | 64 | 58.6 | 69.4 | 25.3 | 39.7 | 12.2 | 38.2 | 36.9 | 40.0 |
| | 128 | 59.3 | 69.4 | 25.3 | 39.1 | 14.4 | 37.0 | 38.3 | 40.4 |

Table 10: **Ablation studies on the effects of rollouts from different models.** The overall Best-of-8 performance is calculated as the average Best-of-8 score across MMMU, MathVista, MathVision, MathVerse-VO, DynaMath, WeMath, and LogicVista.

| Dataset | Overall Best-of-8 Performance | VisualProcessBench Acc. |
|---|---|---|
| VisualPRM400K | 41.2 | 62.0 |
| +44K rollouts from Qwen2.5-VL | 40.9 | 61.5 |
| +30K rollouts from MiMo-VL | 41.8 | 62.7 |
| +74K rollouts from Qwen-2.5-VL and MiMo-VL | 41.9 | 63.1 |

## C  MORE EXPERIMENTAL RESULTS

### C.1  EVALUATION RESULTS BEYOND REASONING BENCHMARKS

As shown in Table 12, we additionally report the model's BoN performance on MMBench, MMStar (perception ability), ChartQA and AI2D (chart understanding), and OCRBench (OCR capability), using the same experimental setup as in Table 2. The results show that VisualPRM-based BoN consistently yields performance improvements, with substantially larger gains than Self-Consistency. These findings provide further evidence of VisualPRM's potential beyond pure reasoning tasks.

### C.2  LATENCY AND THROUGHPUT

Table 13 reports the average per-response latency, which measures the overall latency of generating eight response candidates under the Best-of-8 setting and producing the final response. Since VisualPRM computes step-level scores in parallel (i.e., using "+" as a placeholder and deriving its score from the output probability, requiring only one forward pass to obtain scores for all steps), it does not introduce significant extra latency compared with Self-Consistency. The dominant latency in the overall inference pipeline still comes from the model's autoregressive generation process.

Notably, we use lmdeploy to sample response candidates from the policy model and transformers to evaluate step-level process rewards with VisualPRM. Compared with Self-Consistency, the VisualPRM-based Best-of-N evaluation adds only the latency of scoring responses. Intuitively, the additional latency introduced by VisualPRM is roughly comparable to the latency required for the policy model (of the same size) to generate the last token of the response.

Table 11: **Ablation studies on the effects of maximum number of steps.** The overall Best-of-8 performance is calculated as the average Best-of-8 score across MMMU, MathVista, MathVision, MathVerse-VO, DynaMath, WeMath, and LogicVista.

| Settings | Overall Best-of-8 Performance | VisualProcessBench Acc. |
|---|---|---|
| max steps=1 | 39.7 | 58.7 |
| max steps=3 | 40.3 | 60.0 |
| max steps=6 | 40.7 | 61.5 |
| max steps=9 | 41.2 | 61.7 |
| max steps=12 | 41.2 | 62.0 |

Table 12: **Overall Best-of-8 results on perception-related benchmarks.**

| Model | MMBench | MMstar | ChartQA | AI2D | OCRBench | Overall |
|---|---|---|---|---|---|---|
| MiniCPM-V2.6-8B | 78.0 | 57.5 | 82.4 | 82.1 | 85.2 | 77.0 |
| w. Self-Consistency | 78.3 | 59.2 | 84.0 | 83.3 | 86.4 | 78.2 |
| w. VisualPRM | 80.3 | 62.1 | 85.0 | 84.3 | 87.0 | 79.7 |
| Qwen2.5-VL-7B | 82.6 | 63.9 | 87.3 | 83.9 | 86.4 | 80.8 |
| w. Self-Consistency | 82.5 | 64.2 | 87.7 | 84.2 | 86.4 | 81.0 |
| w. VisualPRM | 83.7 | 64.8 | 87.9 | 85.0 | 86.9 | 81.7 |
| InternVL2.5-8B | 83.2 | 62.8 | 84.8 | 84.5 | 82.2 | 79.5 |
| w. Self-Consistency | 84.4 | 64.5 | 85.6 | 84.9 | 82.5 | 80.4 |
| w. VisualPRM | 85.0 | 65.4 | 85.9 | 86.5 | 87.0 | 82.0 |
| InternVL2.5-78B | 87.4 | 69.5 | 88.3 | 89.1 | 85.4 | 83.9 |
| w. Self-Consistency | 87.2 | 70.3 | 89.9 | 89.5 | 86.0 | 84.6 |
| w. VisualPRM | 87.5 | 72.0 | 89.7 | 89.7 | 90.3 | 85.8 |

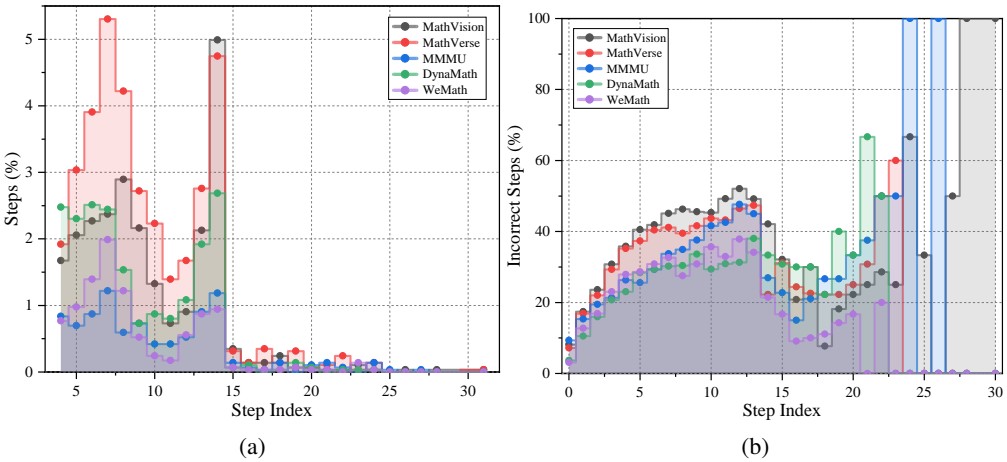

(a)    (b)

Figure 5: **Step Distribution of VisualProcessBench.** The X-axis represents the step index. (a) The Y-axis indicates the proportion of steps at each index relative to the total number of steps, reflecting the distribution of step positions in solutions. (b) The Y-axis represents the error rate of steps at each index, showing the likelihood of errors occurring at different step positions.

## D   MORE STATISTICS FOR VISUALPROCESSBENCH

The statistics for step distribution of VisualProcessBench is presented in Figure 5. We observe that most solutions consist of fewer than 15 steps. Among these solutions with fewer than 15 steps, most solutions contain about 7 or 13 steps. For the correctness of each step, we observe that the error rate is lower in the first three steps and then increases as the step index grows. We attribute this to the fact that problems requiring more reasoning steps tend to be more challenging, leading to a gradual rise in step error rates. Notably, starting from step 15, the error rate drops sharply. This is because the number of steps in this range is relatively small, resulting in significant statistical fluctuations.

Table 13: **The overall Best-of-8 performance and average per-response latency of Self-Consistency and VisualPRM.**

| Model | Overall Best-of-8 Performance | Latency per response (s) |
|---|---|---|
| InternVL2.5-8B-SC | 38.8 | 7.51 |
| InternVL2.5-8B-VisualPRM | 41.2 | 7.76 |
| InternVL2.5-38B-SC | 47.0 | 20.00 |
| InternVL2.5-38B-VisualPRM | 50.7 | 20.26 |
| InternVL2.5-78B-SC | 49.2 | 27.94 |
| InternVL2.5-78B-VisualPRM | 51.9 | 28.33 |

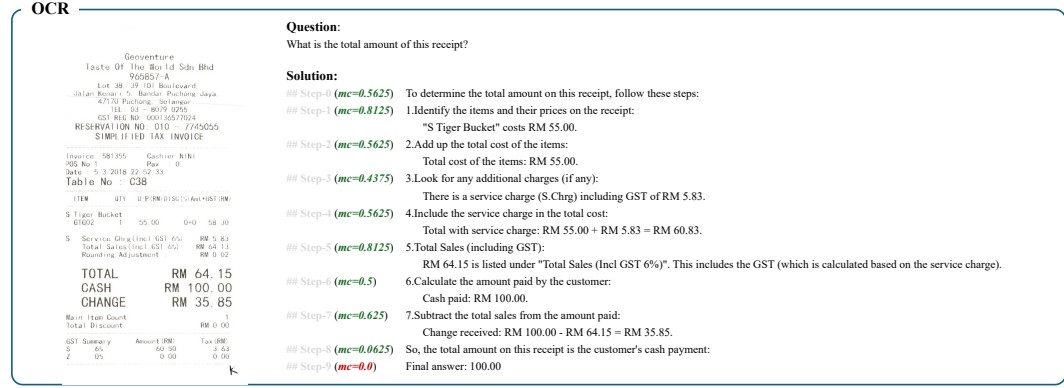

(a)

(b)

## E   MORE DATA EXAMPLES IN VISUALPRM400K

In this section, we provide more data examples of VisualPRM400K in Figure 6 from different domains, including general visual question answering (VQA) (Goyal et al., 2017; Hudson & Manning, 2019; Marino et al., 2019; Lu et al., 2021b), science (Kembhavi et al., 2016; Chen et al., 2024a; Lu et al., 2022), chart (Masry et al., 2022; Kafle et al., 2018; Chang et al., 2022), mathematics (Lindström & Abraham, 2022; Seo et al., 2015; Cao & Xiao, 2022; Lu et al., 2021a; Kazemi et al., 2023; Gao et al., 2023), OCR (Mathew et al., 2022; Singh et al., 2019; Biten et al., 2019; Huang et al., 2019; Mishra et al., 2019), and document (Clark & Gardner, 2018).

## F   MORE DATA EXAMPLES IN VISUALPROCESSBENCH

In this section, we provide more data examples in VisualProcessBench from different data sources. Specifically, we randomly choose three examples from our benchmark and visualize them in Figure 7. Additionally, in Figure 8, we provide an example where the model initially generates an incorrect answer and then autonomously corrects it.

## G   LARGE LANGUAGE MODEL USAGE.

In this submission, we utilize LLMs (*e.g.*, GPT-5) to help us polish paper writing.

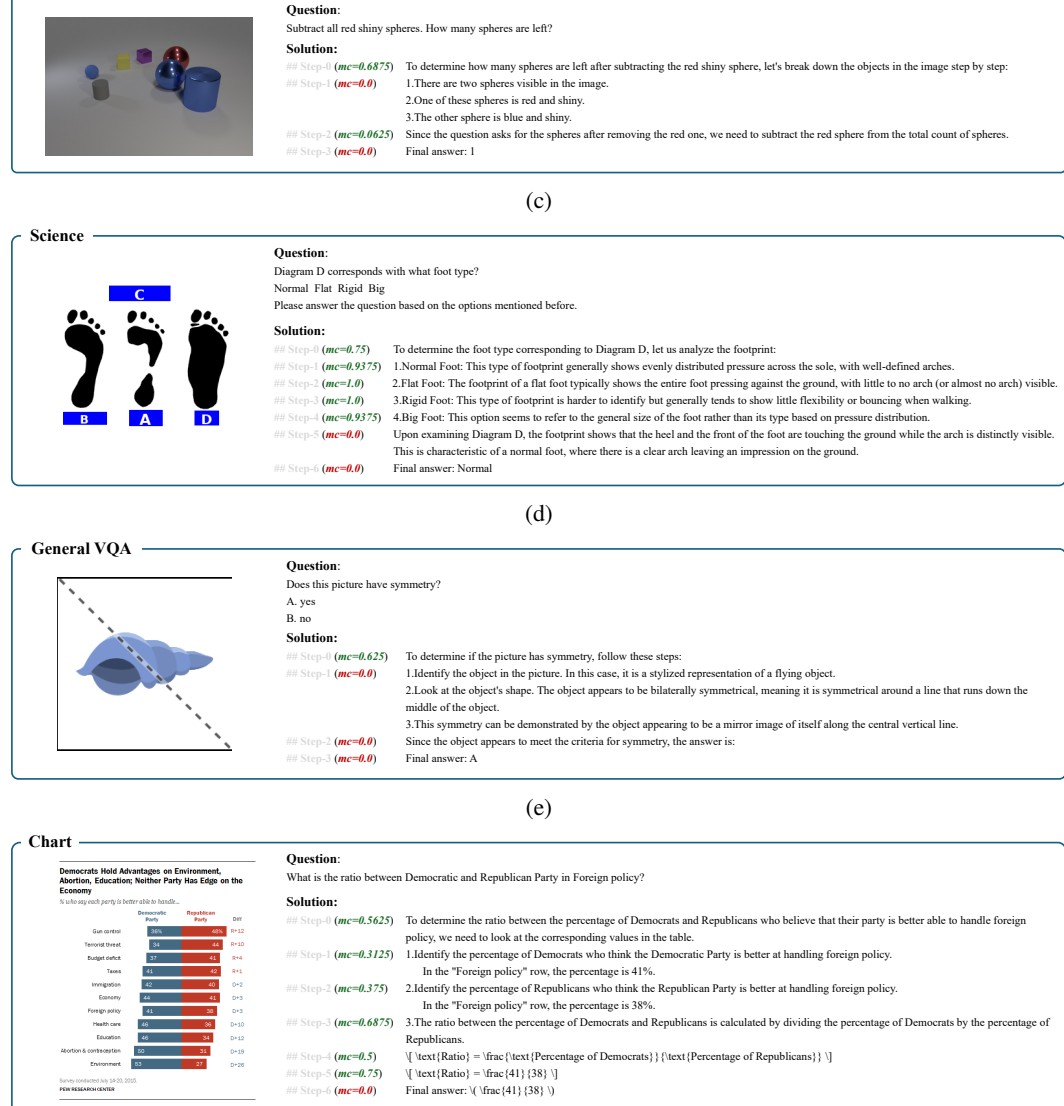

Figure 6: **More data examples from VisualPRM400K.**

Data Example from MMMU

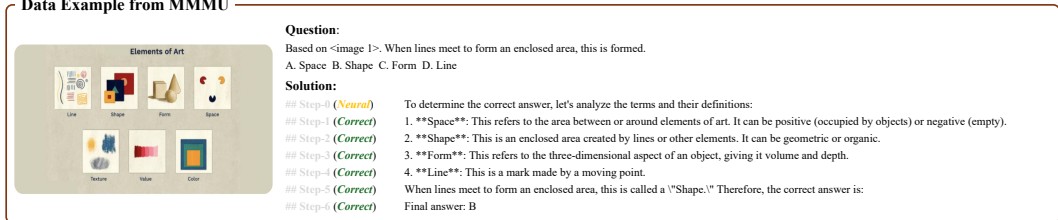

(a)

Data Example from MMMU

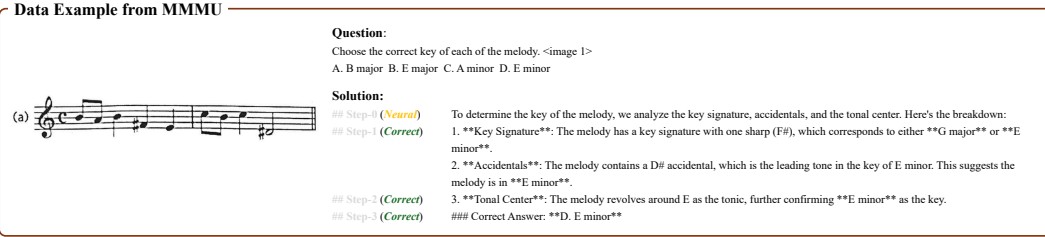

(b)

Data Example from MMMU

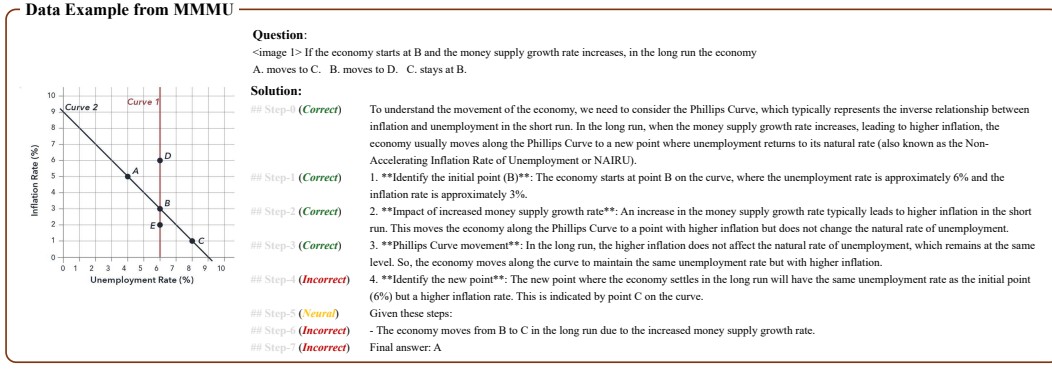

(c)

Data Example from MathVision

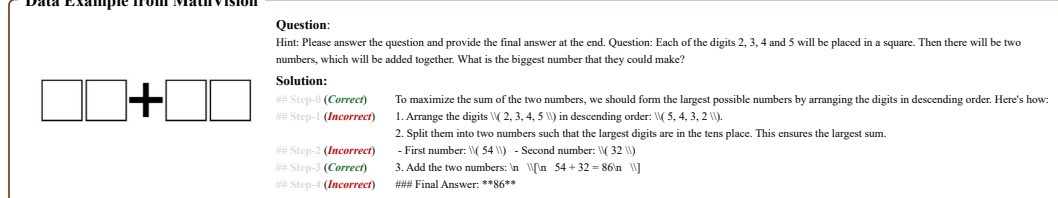

(d)

**Data Example from MathVision**

**Question**:

Hint: Please answer the question and provide the correct option letter, e.g., A, B, C, D, at the end.

Question: Four of the following five pictures show pieces of the graph of the same quadratic function. Which piece does not belong?

Choices:(A) A  (B) B  (C) C  (D) D  (E) E

**Solution:**

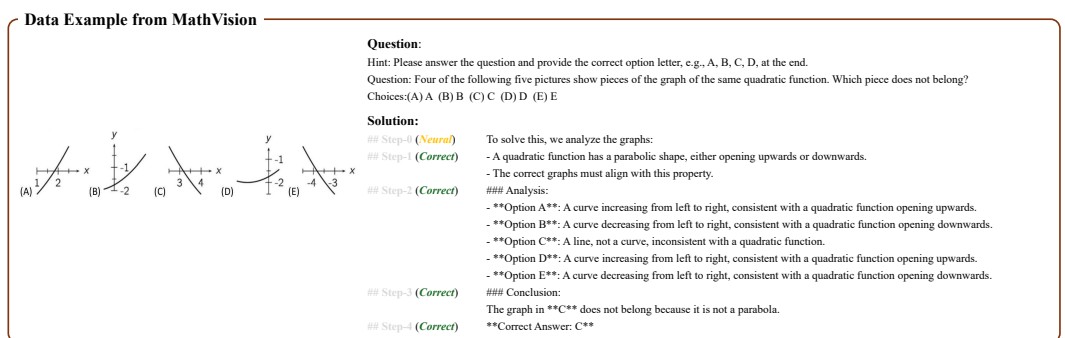

## Step-0 (*Neural*)  To solve this, we analyze the graphs:

## Step-1 (*Correct*)  - A quadratic function has a parabolic shape, either opening upwards or downwards.
- The correct graphs must align with this property.

## Step-2 (*Correct*)  ### Analysis:
- **Option A**: A curve increasing from left to right, consistent with a quadratic function opening upwards.
- **Option B**: A curve decreasing from left to right, consistent with a quadratic function opening downwards.
- **Option C**: A line, not a curve, inconsistent with a quadratic function.
- **Option D**: A curve increasing from left to right, consistent with a quadratic function opening upwards.
- **Option E**: A curve decreasing from left to right, consistent with a quadratic function opening downwards.

## Step-3 (*Correct*)  ### Conclusion:
The graph in **C** does not belong because it is not a parabola.

## Step-4 (*Correct*)  **Correct Answer: C**

(e)

**Data Example from MathVision**

**Question**:

Hint: Please answer the question and provide the final answer at the end.

Question: How many fish will have their heads pointing towards the ring when we straighten the line?

**Solution:**

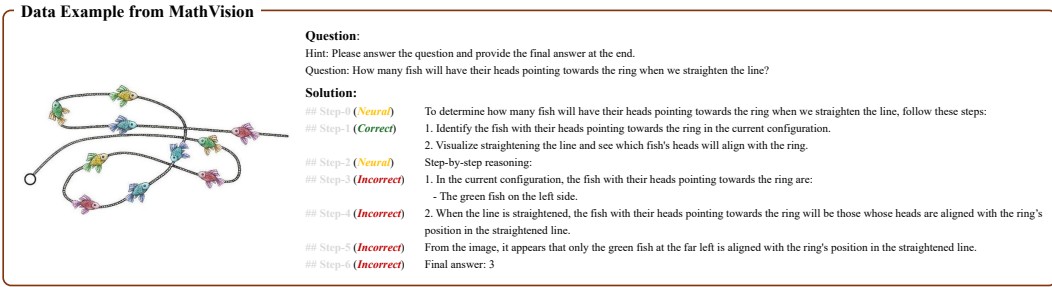

## Step-0 (*Neural*)  To determine how many fish will have their heads pointing towards the ring when we straighten the line, follow these steps:

## Step-1 (*Correct*)  1. Identify the fish with their heads pointing towards the ring in the current configuration.
2. Visualize straightening the line and see which fish's heads will align with the ring.

## Step-2 (*Neural*)  Step-by-step reasoning:

## Step-3 (*Incorrect*)  1. In the current configuration, the fish with their heads pointing towards the ring are:
  - The green fish on the left side.

## Step-4 (*Incorrect*)  2. When the line is straightened, the fish with their heads pointing towards the ring will be those whose heads are aligned with the ring's position in the straightened line.

## Step-5 (*Incorrect*)  From the image, it appears that only the green fish at the far left is aligned with the ring's position in the straightened line.

## Step-6 (*Incorrect*)  Final answer: 3

(f)

**Data Example from MathVerse**

**Question**:

According to the question shown in the image, please directly answer the question and provide the final value, e.g., 1, 2.5, 300.

**Solution:**

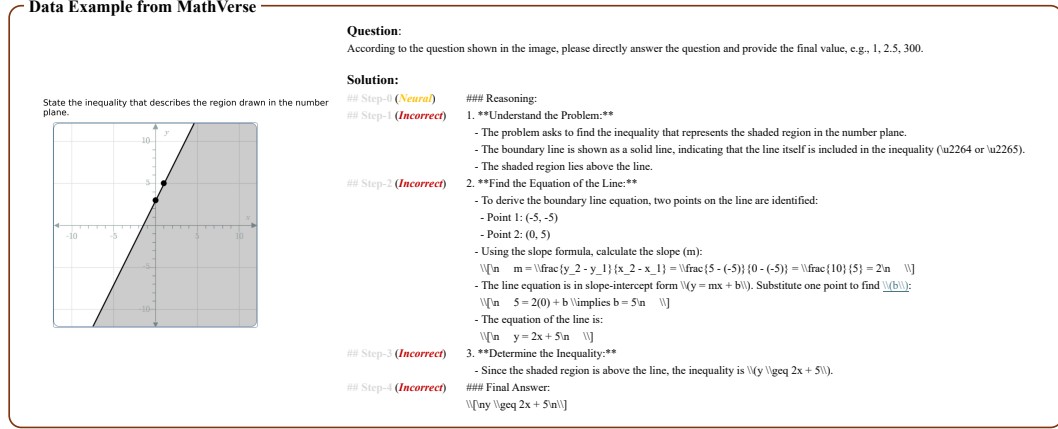

## Step-0 (*Neural*)  ### Reasoning:

## Step-1 (*Incorrect*)  1. **Understand the Problem:**
  - The problem asks to find the inequality that represents the shaded region in the number plane.
  - The boundary line is shown as a solid line, indicating that the line itself is included in the inequality (\u2264 or \u2265).
  - The shaded region lies above the line.

## Step-2 (*Incorrect*)  2. **Find the Equation of the Line:**
  - To derive the boundary line equation, two points on the line are identified:
    - Point 1: (-5, -5)
    - Point 2: (0, 5)
  - Using the slope formula, calculate the slope (m):
    $$ m = \frac{y_2 - y_1}{x_2 - x_1} = \frac{5 - (-5)}{0 - (-5)} = \frac{10}{5} = 2 $$
  - The line equation is in slope-intercept form $(y = mx + b)$. Substitute one point to find $(b)$:
    $$ 5 = 2(0) + b \implies b = 5 $$
  - The equation of the line is:
    $$ y = 2x + 5 $$

## Step-3 (*Incorrect*)  3. **Determine the Inequality:**
  - Since the shaded region is above the line, the inequality is $(y \geq 2x + 5)$.

## Step-4 (*Incorrect*)  ### Final Answer:
$$ y \geq 2x + 5 $$

(g)

**Data Example from MathVerse**

**Question**:

According to the question shown in the image, please directly answer the question and provide the correct option letter, e.g., A, B, C, D.

As shown in the figure, passing point C to draw the tangent of circle O. then the degree of angle A is ()
Choices: A:20°  B:25°  C:.40°  D:50°

**Solution:**

## Step-0 (*Correct*) — To find the degree of angle A, we can use the property that the angle subtended by an arc at the center of the circle is twice the angle subtended at any point on the circle.

## Step-1 (*Correct*) — 1. **Identify the key points:**
- The arcs are AC and BC.
- Angle A is subtended by these arcs at point O (the center of the circle).

## Step-2 (*Correct*) — 2. **Understand the given information:**
- CD is a tangent at point C.
- The angle between the tangent and the chord is equal to the angle subtended by the chord at the circumference. Therefore, angle CDB = 50\u00b0.

## Step-3 (*Incorrect*) — 3. **Angle properties:**
- The angle A is subtended by arc BC at the center.
- Since CD is a tangent, angle A = 50\u00b0 (as the inscribed angle is half the central angle).

## Step-4 (*Correct*) — Thus, the degree of angle A is 20\u00b0.

## Step-5 (*Correct*) — Final answer: A

(h)

**Data Example from MathVerse**

**Question**:

According to the question shown in the image, please directly answer the question and provide the correct option letter, e.g., A, B, C, D.

$f(x)=x^2+x-6$. How many roots do the functions have in common, f(x) and g(x)?
Choices: A:$f$ and $g$ share the same root(s).
B:$f$ and $g$ share one root in common but each have another root that is not shared.
C:$f$ and $g$ share no roots in common.

**Solution:**

## Step-0 (*Neural*) — ### Reasoning:

## Step-1 (*Correct*) — 1. **Roots of \\( f(x) \\):**
The given function is \\( f(x) = x^2 + x - 6 \\), a quadratic equation.
To find its roots, factorize it:
\\[\n  x^2 + x - 6 = (x + 3)(x - 2)\n  \\]
So, the roots of \\( f(x) \\) are \\( x = -3 \\) and \\( x = 2 \\).

## Step-2 (*Incorrect*) — 2. **Roots of \\( g(x) \\):**
Observing the graph of \\( g(x) \\), the parabola intersects the x-axis at \\( x = -3 \\) and \\( x = 4 \\).
Thus, the roots of \\( g(x) \\) are \\( x = -3 \\) and \\( x = 4 \\).

## Step-3 (*Incorrect*) — 3. **Comparison of the roots:**
- \\( f(x) \\) has roots \\( -3 \\) and \\( 2 \\).
- \\( g(x) \\) has roots \\( -3 \\) and \\( 4 \\).
- The functions \\( f(x) \\) and \\( g(x) \\) share one root in common, \\( x = -3 \\), but they each have another root that is not shared (\\( x = 2 \\) for \\( f(x) \\) and \\( x = 4 \\) for \\( g(x) \\)).

## Step-4 (*Incorrect*) — ### Answer:
The correct option is **B**. \\( f \\) and \\( g \\) share one root in common but each have another root that is not shared.

(i)

**Data Example from DynaMath**

**Question**:

Based on the measurement results shown in the diagram, this blue angle is ( )\u00b0.

**Solution:**

## Step-0 (*Neural*) — To determine the measure of the blue angle in the diagram, follow these steps:

## Step-1 (*Incorrect*) — 1. Observe the protractor: The protractor measures angles from 0\u00b0 to 180\u00b0.
2. Identify the measurement: The blue angle is indicated by a line pointing to a specific degree on the protractor.
3. Locate the angle: The blue line points to the number 40\u00b0 on the protractor.

## Step-2 (*Incorrect*) — Thus, the measure of the blue angle is 40\u00b0.

## Step-3 (*Incorrect*) — Final answer: 40

(j)

**Data Example from DynaMath**

| city | Sales number |
|---|---|
| Briggs Corner | 83 |
| New Humburg | 83 |
| Melville | 86 |
| Fremont | 81 |
| liberty City | 92 |
| Charles Falls | 84 |
| Pleasent Town | 93 |
| Penny Town | 97 |

**Question**:

A real estate agent looked into how many houses were sold in different cities. What is the range of the numbers?

**Solution:**

## Step-0 (*Neural*) — Let me help you find the range of the sales numbers.

## Step-1 (*Incorrect*) — The range is calculated by subtracting the minimum value from the maximum value.

## Step-2 (*Incorrect*) — Minimum sales number: 81 (Fremont)\nMaximum sales number: 97 (Penny Town)

## Step-3 (*Incorrect*) — Range = Maximum - Minimum\nRange = 97 - 81 = 16

## Step-4 (*Incorrect*) — Therefore, the range of house sales numbers across these cities is 16.

(k)

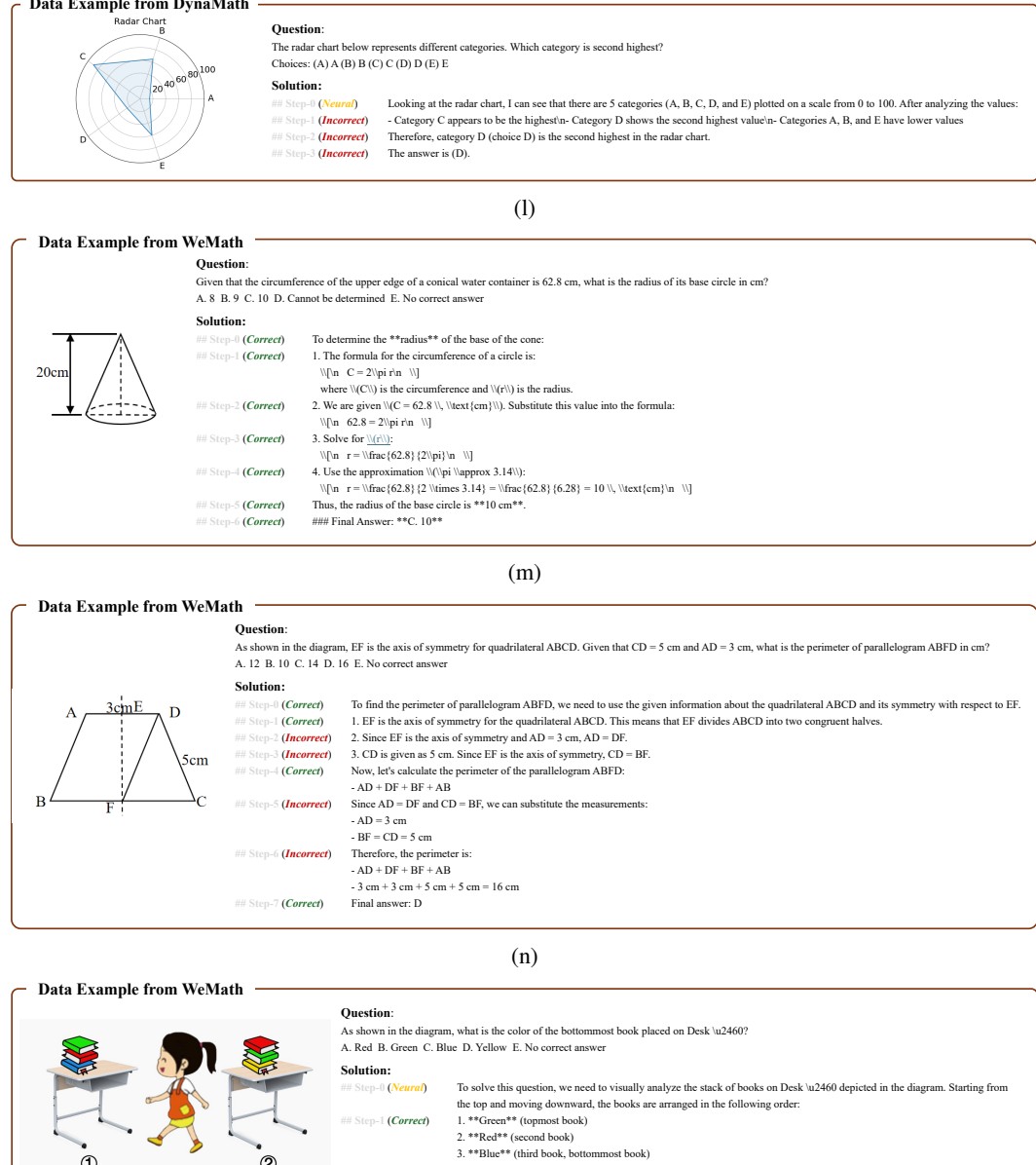

Figure 7: **More data examples from VisualProcessBench.**

**Data Example from VisualProcessBench**

**Question**:
As shown in the figure, triangle ABC is an isosceles triangle, ∠B = 25°, then the angle of ∠1 is ( )
A. 95°  B. 145°  C. 120°  D.  No correct answer

**Solution:**

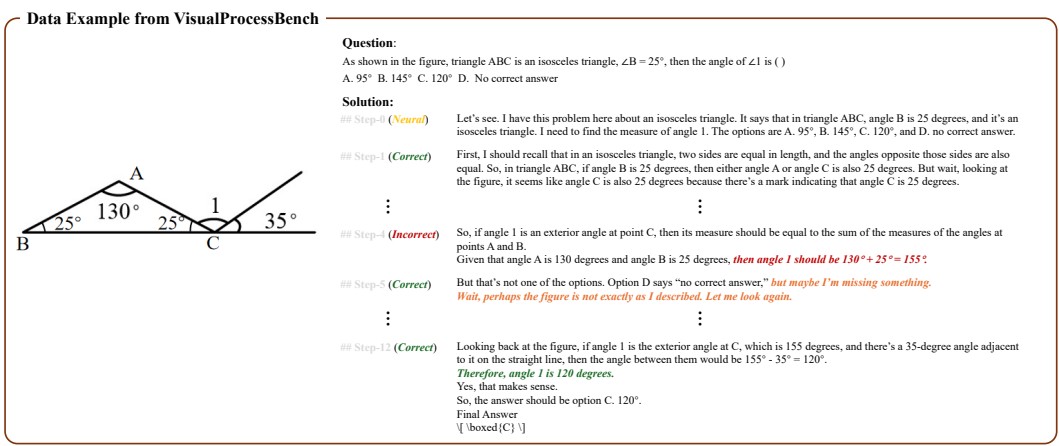

## Step-0 (*Neural*)
Let's see. I have this problem here about an isosceles triangle. It says that in triangle ABC, angle B is 25 degrees, and it's an isosceles triangle. I need to find the measure of angle 1. The options are A. 95°, B. 145°, C. 120°, and D. no correct answer.

## Step-1 (*Correct*)
First, I should recall that in an isosceles triangle, two sides are equal in length, and the angles opposite those sides are also equal. So, in triangle ABC, if angle B is 25 degrees, then either angle A or angle C is also 25 degrees. But wait, looking at the figure, it seems like angle C is also 25 degrees because there's a mark indicating that angle C is 25 degrees.

⋮

## Step-4 (*Incorrect*)
So, if angle 1 is an exterior angle at point C, then its measure should be equal to the sum of the measures of the angles at points A and B.
Given that angle A is 130 degrees and angle B is 25 degrees, *then angle 1 should be 130° + 25° = 155°.*

## Step-5 (*Correct*)
But that's not one of the options. Option D says "no correct answer," *but maybe I'm missing something.*
*Wait, perhaps the figure is not exactly as I described. Let me look again.*

⋮

## Step-12 (*Correct*)
Looking back at the figure, if angle 1 is the exterior angle at C, which is 155 degrees, and there's a 35-degree angle adjacent to it on the straight line, then the angle between them would be 155° - 35° = 120°.
*Therefore, angle 1 is 120 degrees.*
Yes, that makes sense.
So, the answer should be option C. 120°.
Final Answer
\[ \boxed{C} \]

Figure 8: **Data example with model reflection from VisualProcessBench.** Red highlights the incorrect answer, orange highlights the reflection words, and green highlights the correct answer.

