# OpenReview forum: "VisualPRM400K: An Effective Dataset for Training Multimodal Process Reward Models"
_ICLR.cc/2026/Conference — ICLR 2026 Poster_

### Official Review · Reviewer_DRVz · 2025-10-17

**Soundness:** 3
**Presentation:** 3
**Contribution:** 2
**Rating:** 4
**Confidence:** 4

**Summary:**

This paper presents three significant contributions to the field of multimodal reasoning. First, it introduces VisualPRM400K, a large-scale dataset of approximately 400,000 samples with step-by-step process supervision, created using an automated data pipeline based on Monte Carlo sampling. Second, leveraging this dataset, the authors train VisualPRM, an 8B parameter Process Reward Model (PRM) designed to evaluate each step of a multimodal reasoning chain. Third, to facilitate the evaluation of such critic models, the paper proposes VisualProcessBench, a new human-annotated benchmark for identifying all incorrect steps within a reasoning process. The authors demonstrate that using VisualPRM as a critic in a Best-of-N (BoN) inference setting consistently improves the performance of various Multimodal Large Language Models (MLLMs) across seven reasoning benchmarks, outperforming Outcome Reward Models (ORMs) and Self-Consistency.

**Strengths:**

1. Significant and High-Quality Data Contribution: The primary strength of this work lies in the creation and release of two valuable resources: the large-scale VisualPRM400K training dataset and the high-quality, human-annotated VisualProcessBench. Constructing such resources is a laborious but crucial endeavor for the community. VisualPRM400K is, to my knowledge, the first large-scale dataset for training multimodal PRMs, and VisualProcessBench provides a much-needed, fine-grained benchmark for evaluating them. These resources will undoubtedly catalyze future research in multimodal reward modeling and reasoning.
2. Thorough and Rigorous Experimentation: The authors have conducted an extensive set of experiments to validate their contributions. They demonstrate the effectiveness of VisualPRM across multiple model families (MiniCPM, Qwen, InternVL) and scales (from 8B to 78B). The ablation studies are comprehensive, comparing PRM with ORM and Self-Consistency, and analyzing the impact of various hyperparameters. The evaluation on VisualProcessBench, which shows that VisualPRM is competitive with powerful proprietary models like Gemini-2.0-Flash, further solidifies the quality of the trained reward model.
3. Well-Written and Clearly Presented: The paper is exceptionally well-organized and clearly written. The motivation, methodology, and results are presented in a logical and easy-to-follow manner, making the paper's contributions accessible and understandable.

**Weaknesses:**

1. Performance Gains Comparison: The central application of VisualPRM is to improve MLLM reasoning via Best-of-N (BoN) inference. While the reported gains are consistent (e.g., +5.9 points for InternVL2.5-78B), the final performance often falls short of what has been achieved by other contemporary methods that focus on improving the policy model itself through advanced training techniques.
2. In contrast, the approach in this paper is purely an inference-time strategy. While it successfully lifts the performance of existing models, it does not fundamentally enhance the models' intrinsic reasoning capabilities. The resulting performance, while improved, does not appear to push the state-of-the-art boundary as significantly as these training-focused methods.
3. High and Under-discussed Inference Cost: The Best-of-N strategy is notoriously expensive. Using BoN with N=8, as is the default in this paper, multiplies the inference cost (both latency and compute) by at least a factor of 8, plus the overhead of running the VisualPRM critic. This makes the method impractical for many real-world applications. While the authors demonstrate performance scaling up to N=128, they do not provide a thorough discussion on the cost-performance trade-off. A more complete analysis would be necessary to assess the practical viability of this approach. The reported performance gains, while notable, may not be sufficient to justify such a substantial increase in inference cost for many use cases.

**Questions:**

1. Could you provide a more direct comparison of your final BoN results with other state-of-the-art methods on the same benchmarks? How does the performance of, for example, "InternVL2.5-78B + VisualPRM" compare to models that have been fine-tuned with advanced RL or self-improvement techniques? This would help contextualize the significance of the improvements you've achieved.
2. Could you elaborate on the inference latency and computational cost of using VisualPRM in a BoN setting? For instance, what is the wall-clock time required to evaluate a single instance with N=8 compared to a single pass from the base model? A cost-benefit analysis would be extremely valuable for readers to understand the practical implications of your method.

---

> ### Author Response · Authors · 2025-11-30
>
> Thank you for your insightful feedback and constructive suggestions.
>
> **Q1: Comparison with methods focusing on training the policy model**
>
> Thank you for the suggestion. ***We emphasize that our VisualPRM-based Best-of-N (BoN) approach is not in conflict with methods that rely on policy model finetuning—the performance gains from the two approaches are complementary and additive.*** Specifically, our BoN method operates during the inference stage, whereas finetuning-based methods improve the model during the training stage.
>
> In the table below, we report the performance gains of InternVL3.5 and InternVL3.5-Inst under the VisualPRM-based BoN setting. Notably, InternVL3.5 is initialized from InternVL3.5-Inst and then finetuned with RL algorithms, while InternVL3.5-Inst is not. The results show that VisualPRM still brings substantial improvements to both models.
>
> |Model|MMMU|MathVista|MathVision|MathVerse-VO|DynaMath|WeMath|LogicVista|Overall|
> |-|-|-|-|-|-|-|-|-|
> |InternVL3.5-8B-Inst|68.1|74.2|46.4|55.8|30.7|46.0|53.9|53.6|
> |w.VisualPRM|71.0|76.3|52.9|63.8|38.0|50.3|57.8|58.6|
> |InternVL3.5-14B-Inst|71.8|73.4|48.7|55.5|31.9|45.7|57.5|54.9|
> |w.VisualPRM|73.0|77.4|54.8|64.7|39.3|53.4|60.2|60.4|
> |InternVL3.5-38B-Inst|73.9|75.9|58.2|59.0|29.7|47.5|60.0|57.7|
> |w.VisualPRM|75.6|79.2|63.3|69.3|37.7|55.5|65.3|63.7|
> |InternVL3.5-241B-Inst|76.2|80.1|55.6|61.7|36.5|49.7|63.3|60.4|
> |w.VisualPRM|77.7|83.8|60.0|71.4|43.4|56.6|67.0|65.7|
>
>
> |Model|MMMU|MathVista|MathVision|MathVerse-VO|DynaMath|WeMath|LogicVista|Overall|
> |-|-|-|-|-|-|-|-|-|
> |InternVL3.5-8B|73.4|78.4|56.8|61.5|37.7|57.0|57.3|60.3|
> |w.VisualPRM|73.4|80.8|59.9|62.6|39.9|57.6|62.6|62.4|
> |InternVL3.5-14B|73.3|80.5|59.9|62.8|38.7|58.7|60.2|62.0|
> |w.VisualPRM|74.3|81.5|61.4|64.1|41.3|60.8|61.3|63.5|
> |InternVL3.5-38B|76.9|81.9|63.7|67.6|41.7|64.8|65.3|66.0|
> |w.VisualPRM|76.7|83.8|65.6|66.5|44.9|67.8|64.7|67.1|
> |InternVL3.5-241B|77.7|82.7|63.9|68.5|46.5|62.3|66.7|66.9|
> |w.VisualPRM|78.7|84.8|65.9|71.6|47.8|64.4|67.6|68.7|
>
>
> Importantly, the performance gains from BoN are smaller for InternVL3.5-Inst than for InternVL3.5. We attribute this to two factors:
> 1. After RL training, the model’s outputs become more consistent, causing Pass@K to converge toward Pass@N [1], thus reducing the relative benefit of sampling-based strategies.
> 2. RL finetuning already strengthens the model’s reasoning ability, leaving less headroom for further improvements.
>
> [1] *Does Reinforcement Learning Really Incentivize Reasoning Capacity in LLMs Beyond the Base Model?*
>
>
>
> **Q2: The inference cost of the BoN strategy**
>
> ***The core objective of Test-Time Scaling is to improve model performance by increasing inference-time computation, which is fundamentally different from and not in conflict with improving performance during training. Increasing inference cost is an intrinsic characteristic of the Test-Time Scaling paradigm.*** Under the Best-of-N setting, users can flexibly adjust N to balance the performance–latency tradeoff.
> Importantly, although Test-Time Scaling introduces higher inference cost, many real-world scenarios value performance over latency, making Test-Time Scaling a valuable research direction.
>
> In the table below, we report the average per-response latency.
> Because VisualPRM computes step-level scores in parallel—using “+” as a placeholder and obtaining scores from its output probabilities with a single forward pass—it does not introduce significant latency compared with Self-Consistency. The dominant cost in the overall inference pipeline continues to be the autoregressive generation by the policy model.
> Intuitively, the additional latency introduced by VisualPRM is approximately equal to the latency required for a policy model of the same size to generate the final token of a response.
>
> |Model| Best-of-8 | Latency per response (s) |
> |-|-|-|
> |InternVL2.5-8B-SC| 38.8| 7.51|
> |InternVL2.5-8B-VisualPRM  | 41.2| 7.76|
> |InternVL2.5-38B-SC| 47.0| 20.00|
> |InternVL2.5-38B-VisualPRM | 50.7| 20.26|
> |InternVL2.5-78B-SC| 49.2| 27.94|
> |InternVL2.5-78B-VisualPRM | 51.9| 28.33|
>
> We also emphasize that although the Best-of-N setting increases inference cost by a factor of  N compared to Pass@1, this behavior is inherent to Best-of-N and Test-Time Scaling itself. The purpose of this setting is explicitly to improve performance by increasing inference computation—therefore, this characteristic should not be viewed as a weakness of our method.

---

### Official Review · Reviewer_edvQ · 2025-10-31

**Soundness:** 3
**Presentation:** 3
**Contribution:** 4
**Rating:** 8
**Confidence:** 4

**Summary:**

The paper introduces VisualPRM400K, a ~400K multimodal process-supervision dataset built via Monte-Carlo (MC) completion to estimate step-wise correctness; an 8B VisualPRM trained on it; and VisualProcessBench, a PRM/MLLM benchmark with 2,866 samples and 26,950 human-annotated step labels. With Best-of-N (BoN) test-time scaling, VisualPRM substantially improves multiple MLLMs (e.g., +8.4 for InternVL2.5-8B; +5.9 for InternVL2.5-78B) and outperforms outcome reward models and self-consistency as the critic.

**Strengths:**

Timely, practical contribution: A large multimodal process dataset plus a purpose-built benchmark for PRMs addresses a clear gap and enables systematic progress on multimodal TTS.

Solid empirical evidence: Consistent BoN gains across model scales; clear comparisons vs. ORM and self-consistency; ablations on value- vs advantage-based PRMs and score aggregation.

Clear PRM formulation: Step-wise discretized targets, single-pass scoring efficiency, and supervising all steps rather than stopping at first error are well motivated and validated.

**Weaknesses:**

Potential generator-bias in labels: Process rewards are derived from continuations sampled with InternVL-2.5 models. This may bias the PRM toward InternVL-style reasoning and limit transfer to other families (e.g., GPT-5, Qwen-VL).

MC estimation only: The paper uses MC to estimate expected accuracy per step but does not compare with alternative credit assignment/judging strategies (e.g., MCTS, LLM-as-a-Judge).

Model size choice and scaling law: VisualPRM is fixed at 8B; the paper lacks justification for this size and a scaling curve (e.g., 1B/3B/8B/14B) to reveal accuracy–latency–cost trade-offs.

**Questions:**

On process reward generation: You estimate step values via MC sampling. Have you tried other strategies such as MCTS rollouts or LLM-as-a-Judge?

On PRM capacity: Why 8B? Have tried other PRM size (e.g., 1B/2B/9B/14B)?

Related work coverage (multimodal PRMs): The Related Work section focuses primarily on text-only PRMs. It should discuss recent multimodal PRM papers such as DreamPRM [1], AR-MCTS [2].

[1] DreamPRM: Domain-Reweighted Process Reward Model for Multimodal Reasoning (NeurIPS 2025)
[2] Progressive Multimodal Reasoning via Active Retrieval (ACL 2025)

---

> ### Author Response · Authors · 2025-11-30
>
> Thank you for your time and expertise in the review process.
>
> **Q1: The generalization ability of VisualPRM to other model families**
>
> In Table 2, we report the performance improvements achieved by policy models from different model families under the VisualPRM-based Best-of-N setting. These results demonstrate VisualPRM’s ability to transfer across model families. In addition, the results in Figure 4 and Tables 7/8 further show that VisualPRM consistently yields performance gains across different choices of N for both InternVL and MiniCPM-V.
>
>
> **Q2: More analysis about the data pipeline**
>
> Thank you for the suggestion. Our data pipeline uses MC sampling to evaluate step correctness in order to ensure scalability. In the early stage, we also explored using MCTS-based approaches to achieve more efficient MC sampling. However, we found that under this setting, the sampling process becomes difficult to parallelize, resulting in suboptimal GPU utilization and reduced data construction efficiency. For this reason, we ultimately opted for the MC sampling approach that maximizes GPU utilization. In addition, LLM-as-a-Judger faces similar challenges. Although current state-of-the-art LLMs (typically 72B or larger, often commercial models) may provide strong judging capabilities, the associated computational and monetary costs are extremely high.
>
>
> **Q3: The capacity and scalability of VisualPRM**
>
> We selected the 7B model size because it is a commonly used size for MLLMs. Here, we provide additional experimental results regarding scalability. Based on the results in the table below, we observe that increasing the size of VisualPRM leads to significantly better performance.
>
> | Model            | MMMU  | MathVista | MathVision | MathVerse-VO | DynaMath | WeMath | LogicVista | Overall |
> |------------------|-------|-----------|------------|--------------|----------|--------|------------|---------|
> | InternVL2.5-8B   | 56.2  | 64.5      | 17.0       | 22.8         | 9.4      | 23.5   | 36.0       | 32.8    |
> | w. VisualPRM-8B  | 60.2  | 68.5      | 25.7       | 35.8         | 18.0     | 36.5   | 43.8       | 41.2    |
> | w. VisualPRM-38B | 62.3  | 69.9      | 30.3       | 37.7         | 18.9     | 38.9   | 45.0       | 43.3    |
> | InternVL2.5-38B  | 63.9  | 71.9      | 32.2       | 36.9         | 20.0     | 38.3   | 47.9       | 44.4    |
> | w. VisualPRM-8B  | 69.0  | 73.9      | 35.2       | 46.7         | 30.5     | 46.2   | 53.7       | 50.7    |
> | w. VisualPRM-38B | 70.1  | 75.5      | 38.9       | 48.3         | 33.3     | 48.8   | 55.6       | 52.9    |
>
>
> **Q4: Discussion with recent multimodal PRMs**
>
> Thank you for the suggestion. We will include a discussion and citations of these papers in the revised version.
> Regarding AR-MCTS, this work primarily focuses on proposing a new reasoning paradigm to enhance the policy model’s performance, whereas DreamPRM focuses on new training algorithms to improve multimodal PRM effectiveness and training efficiency.
> In contrast, our work centers on constructing training data and benchmarks for multimodal PRMs.

---

### Official Review · Reviewer_h3nV · 2025-11-01

**Soundness:** 2
**Presentation:** 3
**Contribution:** 2
**Rating:** 2
**Confidence:** 4

**Summary:**

The paper introduces VisualPRM400K, a ~400K-sample multimodal process-supervision dataset with step-wise expected-accuracy labels, and VisualProcessBench, a 2,866-sample benchmark with 26,950 human step-correctness annotations, to enable and evaluate process reward models (PRMs) as critics for Best-of-N test-time scaling in MLLMs. Trained as an 8B PRM that scores each reasoning step in a single forward pass, VisualPRM achieved great improvement across seven multimodal reasoning benchmarks.

**Strengths:**

1. The paper is generally well-written and easy to follow, with a clear description of the method.
2. The paper provides intuitive visual demonstrations to help better understand the paper.

**Weaknesses:**

1. Label quality & construction pipeline clarity. The Monte-Carlo step-correctness (`Eq. (2)`) relies on continuations sampled from an unspecified model $M$; this risks systematic bias/noise if $M$ shares failure modes with the policy models later evaluated. In addition, merging solutions to a maximum of 12 steps may distort error localization and the temporal dynamics of “first error vs. downstream errors.” Please quantify label noise (e.g., step-level inter-rater agreement on a subset), report sensitivity to the number of sampled continuations, and analyze the effect of step-merging on PRM accuracy.

2. Fairness of comparisons. Please clarify whether the reported gains in `Tab.2` are measured only against each policy model’s base (“Pass@1”) or also against strong critic baselines. To more comprehensively validate the effectiveness of the proposed approach, include outcome-based reward (ORM) and additional PRM baselines under identical Best-of-N settings (same candidate pool, N, decoding, and compute). Also expand `Tab.3` with more PRM variants and report matched-compute results.

3. Generalization beyond the current suite. Beyond the seven benchmarks used (six math, one multidisciplinary), consider evaluating on more general-purpose multimodal benchmarks and on broader text-only reasoning benchmarks to substantiate cross-domain robustness.

4. Limited technical novelty; strengthen the case for multimodality. The paper’s primary contributions appear to be `VISUALPRM400K` and `VisualProcessBench`, while methodological novelty is modest. To demonstrate that the *multimodal* aspect is indispensable (rather than a text-dominant signal), please add modality ablations and analyses showing performance drops when visual evidence is removed or corrupted. Such results would clarify the unique value of multimodal supervision/assessment relative to single-modal PRMs.

**Questions:**

See the `Weaknesses` part.

---

> ### Author Response · Authors · 2025-11-30
>
> Thank you for your efforts in the review.
>
> **Q1: More analysis about the data pipeline.**
>
> Thank you for the suggestion. We provide additional analysis in *Common Questions Q2*. Specifically, we found that incorporating more rollouts sampled from policy models as training data helps improve the performance of VisualPRM. In addition, our motivation for setting a maximum number of steps is to reduce data construction costs. In our data pipeline, the construction cost scales linearly with the maximum number of steps. Results in *Common Questions Q2* show that increasing the maximum number of steps in the training data indeed improves performance, but the gains exhibit clear diminishing returns. For example, the improvement from increasing the maximum steps from 10 to 12 is already very marginal. Please refer to *Common Questions Q2* for the detailed experimental results.
>
>
> **Q2: Comparison settings.**
>
> In Table 2, the performance gains highlighted in red represent the improvement of Best-of-8 with VisualPRM compared to the Pass@1 performance of the policy model. In Figure 4, we provide a comparison among Self-Consistency, ORM, and PRM under *exactly the same settings* (identical candidate pool, N, decoding strategy, and compute budget). More detailed results on this part can be found in Tables 7 and 8.
> In Table 4, we further compare different PRM modeling approaches and analyze their impact on BoN performance as well as performance on VisualProcessBench.
>
>
> **Q3: Evaluation results beyond reasoning benchmarks.**
>
> Thank you for the suggestion. We additionally report the model’s BoN performance on MMBench and MMStar (perception tasks), ChartQA and AI2D (chart understanding), and OCRBench (OCR capability). The results show that VisualPRM-based BoN consistently yields performance improvements, with gains larger than those of Self-Consistency. These results provide evidence of VisualPRM’s potential on perception-oriented tasks. Please refer to *Common Questions Q3* for the detailed experimental results.
>
>
> **Q4: The key contributions of this paper.**
>
> We emphasize that the primary contribution and effort of our work lie in the creation of the VisualPRM400K dataset and VisualProcessBench, rather than proposing a novel model architecture or training algorithm. Please refer to *Common Questions Q1* for the detailed response.

---

### Official Review · Reviewer_Ppnx · 2025-11-01

**Soundness:** 3
**Presentation:** 3
**Contribution:** 3
**Rating:** 6
**Confidence:** 3

**Summary:**

The paper introduces a new dataset to train multimodal Process Reward Models, a new benchmark to evaluate MM PRMs and an 8B-parameter PRM that consistently, across different model series and sizes, improves performance over MM reasoning benchmarks. The PRM is shown to outperform other TTS algorithms like Outcome Reward Models, Self-Consistency and MLLMs as critic models.

**Strengths:**

1.	TTS and using PRMs as reward functions in RL are under explored in multimodal modeling. The work here pledges to open source a large 400k sample dataset to train MM-PRMs, a benchmark to evaluate MM-PRMs and a trained MM-PRM. These contributions can be valuable for the community and foster further research.
2.	Effectiveness of PRMs used for TTS in improving MM reasoning across multiple model series and sizes are clearly demonstrated along with improvements compared to other TTS methods.
3.	Design of the PRM benchmark  is sound with the main standouts being, considering all the erroneous reasoning steps as opposed to stopping at the first occurrence, using macro F1 scores to account for class imbalance and multiple math and reasoning datasets.

**Weaknesses:**

1.	PRMs are well studied in the language modeling space. This work repurposes those studies and algorithms to the multimodal space with limited algorithmic novelty.
2.	An automated Monte Carlo sampling-based pipeline is used to generate the PRM400k dataset. There are no discussions on the quality of this dataset and alignment with human judgement. How the authors ensure incorrect demonstrations are filtered out and how this could affect the trained PRM’s abilities as a critic are not discussed. 10% of the reasoning steps are negative demonstrations. Effect of PRM modeling with more balanced positive and negative demonstrations, using weaker models and models from other series to introduce diverse thinking styles is not explored.
3.	The task categories in VisualProcessBench are mostly centered around math and logic. Extensibility of this methodology for other vision applications like chart, table, GUI reasoning among others will be helpful.
4.	While PRM is shown to outperform other Best-of-N strategies, a discussion about latency and throughput tradeoffs compared to other light-weight strategies can strengthen the claim.

**Questions:**

1.	Can the authors quantify FP and FN rates for a sub-sample of the VisualPRM400k dataset and explain if they have any filtering steps to identify and remove such demonstrations?
2.	Can the authors provide accuracy-vs-latency plots at multiple N comparing different BoN techniques?

---

> ### Author Response · Authors · 2025-11-30
>
> Thank you for your valuable feedback and constructive comments.
>
> **Q1: The key contributions of this paper.**
>
> We emphasize that the primary contributions and efforts of our work lie in the construction of the VisualPRM400K dataset and VisualProcessBench, rather than proposing a novel model architecture or training algorithm. Please refer to *Common Questions Q1* for a more detailed response to this point.
>
> **Q2: More analysis about the data pipeline.**
>
> Thank you for the suggestion. We introduced additional PRM training data constructed from rollouts sampled using Qwen2.5-VL-7B and MiMo-VL-7B, and we provide the corresponding Best-of-N (BoN) performance of PRMs trained with these augmented datasets. Due to the time constraints of the rebuttal period, we were able to construct only about 74K new training samples (44K from Qwen2.5-VL and 30K from MiMo-VL-7B). Experimental results show that incorporating more diverse training data sampled from additional models  indeed leads to further performance improvements for VisualPRM. Please refer to *Common Questions Q2* for the detailed results.
>
> **Q3: The evaluation task categories.**
>
> Thank you for the suggestion. Since VisualProcessBench is based on human annotations, we were unable to complete additional data annotation within the rebuttal deadline. Instead, we provide the model’s BoN performance on MMBench and MMStar (perception capability), ChartQA and AI2D (chart understanding capability), and OCRBench (OCR capability). The results demonstrate that BoN with VisualPRM consistently yields higher performance gains than Self-Consistency. These results provide evidence of VisualPRM’s potential in perception-oriented tasks. Please refer to *Common Questions Q3* for the detailed experimental results.
>
> **Q4: More discussion about latency and throughput.**
>
> Thank you for the suggestion. The table below reports the average per-response latency, which measures the overall latency of generating eight response candidates under the Best-of-8 setting and producing the final response. Since VisualPRM computes step-level scores in parallel (i.e., using “+” as a placeholder and deriving its score from the output probability, requiring only **one forward pass** to obtain scores for all steps), it does **not** introduce significant extra latency compared with Self-Consistency. The dominant latency in the overall inference pipeline still comes from the model’s autoregressive generation process.
>
> | Model                     | Best-of-8 | Latency per response (s) |
> |---------------------------|-----------|--------------------------|
> | InternVL2.5-8B-SC         | 38.8      | 7.51                     |
> | InternVL2.5-8B-VisualPRM  | 41.2      | 7.76                     |
> | InternVL2.5-38B-SC        | 47.0      | 20.00                    |
> | InternVL2.5-38B-VisualPRM | 50.7      | 20.26                    |
> | InternVL2.5-78B-SC        | 49.2      | 27.94                    |
> | InternVL2.5-78B-VisualPRM | 51.9      | 28.33                    |
>
> Notably, we use lmdeploy to sample response candidates from the policy model and transformers to evaluate step-level process rewards with VisualPRM. Compared with Self-Consistency, the VisualPRM-based Best-of-N evaluation adds only the latency of scoring responses. Intuitively, the additional latency introduced by VisualPRM is roughly comparable to the latency required for the policy model (of the same size) to generate the last token of the response.

---

### Author Response · Authors · 2025-11-30
**General Response**

Dear all reviewers,

We sincerely appreciate the reviewers for your time and effort in evaluating our work. First, we would like to highlight and clarify the key contributions of our research. We then address the common questions, followed by detailed responses to each reviewer. We hope our explanations will help resolve any concerns.


**Q1: The key contributions of this paper.**

As discussed in L045–L056, a central challenge in developing multimodal PRMs lies in the lack of large-scale training data and the absence of standardized benchmarks. Accordingly, the primary contribution and effort of our work lie in the creation of VisualPRM400K and VisualProcessBench, rather than in proposing a novel model architecture. These resources establish a strong foundation for subsequent research in this domain.

Although the data pipeline itself is relatively simple, constructing VisualPRM400K required over 2,000 A800 GPU hours, primarily due to the computational demands of Monte Carlo sampling for CoT reasoning. By providing these resources, our work reduces the heavy cost of dataset construction and enables consistent evaluation, allowing future studies to focus more directly on *algorithmic* and *methodological* innovation.

Furthermore, the simplicity of our modeling approach is an intentional and practical design choice. To clearly demonstrate the utility of our proposed dataset and benchmark, we adopt a baseline that is simple, effective, and easy to reproduce.

As shown in Section 4, even this simple approach achieves strong results, validating both the quality of our resources and the potential of multimodal PRMs.


**Q2: More analysis about the data pipeline.**

Here we provide additional analysis of the data pipeline. Specifically, we incorporate additional PRM training data constructed from rollouts sampled using Qwen2.5-VL-7B and MiMo-VL-7B, and we report the BoN performance of PRMs trained on this expanded dataset. Due to the time constraints of the rebuttal period, we were able to construct only ~74K additional samples (44K from Qwen2.5-VL and 30K from MiMo-VL-7B). The construction process is identical to that of VisualPRM400K, with the only difference being the policy model used for sampling. Experimental results confirm that incorporating more diverse rollouts from additional models further improves VisualPRM performance.

Notably, "+44K rollouts from Qwen2.5-VL" indicates that the training dataset is constructed by adding an additional 44K samples on top of VisualPRM400K.

|Dataset| Best-of-8 | VisualProcessBench Acc. |
|-|-|-|
|VisualPRM400K| 41.2      | 62.0|
|+44K rollouts from Qwen2.5-VL| 40.9      | 61.5 |
|+30K rollouts from MiMo-VL| 41.8      | 62.7 |
|+74K rollouts from Qwen-2.5-VL and MiMo-VL | 41.9| 63.1|

In addition, we analyze how the maximum number of steps in data construction influences model performance. To reduce annotation cost, we reused the annotations from the max-step = 12 configuration to build datasets with max-step = 1/3/6/9. We then trained separate VisualPRM models on these subsets. As shown in the table below, increasing the maximum step count does improve performance, but the gains exhibit clear diminishing returns—for instance, increasing the max steps from 10 to 12 yields only very marginal improvement.

| Settings| Best-of-8 | VisualProcessBench Acc. |
|-|-|-|
| max steps=1  | 39.7| 58.7 |
| max steps=3  | 40.3| 60.0|
| max steps=6  | 40.7| 61.5|
| max steps=9  | 41.2| 61.7|
| max steps=12 | 41.2| 62.0|

**Q3: Evaluation results beyond reasoning benchmarks.**

As shown in the table below, we additionally report the model’s BoN performance on MMBench, MMStar (perception ability), ChartQA and AI2D (chart understanding), and OCRBench (OCR capability), using the same experimental setup as in Table 2. The results show that VisualPRM-based BoN consistently yields performance improvements, with substantially larger gains than Self-Consistency. These findings provide further evidence of VisualPRM’s potential beyond pure reasoning tasks.

| Model| MMBench | MMstar | ChartQA | AI2D  | OCRBench | Overall |
|-|-|-|-|-|-|-|
| MiniCPM-V2.6-8B | 78.0    | 57.5   | 82.4    | 82.1  | 85.2     | 77.0|
| w. Self-Consistency| 78.3    | 59.2   | 84.0    | 83.3  | 86.4     | 78.2|
| w. VisualPRM| 80.3    | 62.1   | 85.0    | 84.3  | 87.0     | 79.7|
| Qwen2.5-VL-7B| 82.6    | 63.9   | 87.3    | 83.9  | 86.4| 80.8|
| w. Self-Consistency | 82.5    | 64.2   | 87.7    | 84.2  | 86.4     | 81.0    |
| w. VisualPRM| 83.7    | 64.8   | 87.9    | 85.0  | 86.9     | 81.7    |
| InternVL2.5-8B| 83.2    | 62.8   | 84.8    | 84.5  | 82.2     | 79.5    |
| w. Self-Consistency | 84.4    | 64.5   | 85.6    | 84.9  | 82.5     | 80.4    |
| w. VisualPRM| 85.0    | 65.4   | 85.9    | 86.5  | 87.0 | 82.0    |
| InternVL2.5-78B| 87.4    | 69.5| 88.3| 89.1  | 85.4| 83.9    |
| w. Self-Consistency | 87.2| 70.3   | 89.9| 89.5  | 86.0     | 84.6    |
| w. VisualPRM| 87.5| 72.0 | 89.7| 89.7  | 90.3| 85.8    |

---

### Meta-Review · Area_Chair_4JNB · 2026-01-07

**Summary:**

- No comparison with alternative credit assignment strategies or other critic strategies (Ppnx, h3nV, edvQ)
- Potential bias towards InternVL-style reasoning. (h3nV, edvQ)
- Best-of-N not as strong as training the underlying model (DRVz)
- Best-of-N is expensive (Ppnx, DRVz)

**Reviewer Concerns:**

- The authors show that BoN composes with RL post-training, but is less effective on RL-ed models.
- The authors argue that GPU efficiency explains their MC sampling choice.
- The authors remind reviewers that the dataset is the central contribution.

**Reviewer Scores:**

- Ppnx: likely improved based on new results from the authors
- h3nV: likely improved based on new benchmark results
- edvQ: again, likely improved based on new results
- DRVz: no change, the latency numbers and comparisons with RL vs non-RL models seem necessary rather than a meaningful improvement.

---

### Decision · Program_Chairs · 2026-01-26

Accept (Poster)